# Effects of Hydrodilatation at Different Volumes on Adhesive Capsulitis in Phases 1 and 2: Clinical Trial Protocol HYCAFVOL

**DOI:** 10.3390/clinpract15080141

**Published:** 2025-07-26

**Authors:** Javier Muñoz-Paz, Ana Belén Jiménez-Jiménez, Francisco Espinosa-Rueda, Amin Wahab-Albañil, María Nieves Muñoz-Alcaraz, José Peña-Amaro, Fernando Jesús Mayordomo-Riera

**Affiliations:** 1Interlevel Clinical Management Unit of Physical Medicine and Rehabilitation, Reina Sofia University Hospital,-Cordoba and Guadalquivir Health District, 14011 Cordoba, Spain; javier.munoz.paz.sspa@juntadeandalucia.es (J.M.-P.); anab.jimenez.jimenez.sspa@juntadeandalucia.es (A.B.J.-J.); francisco.espinosa.rueda.sspa@juntadeandalucia.es (F.E.-R.); amin.wahab.sspa@juntadeandalucia.es (A.W.-A.); cm1peamj@uco.es (J.P.-A.); sr1marif@uco.es (F.J.M.-R.); 2Maimonides Biomedical Research Institute of Cordoba (IMIBIC), Reina Sofia University Hospital, University of Cordoba, 14004 Cordoba, Spain; 3Department of Applied Physics, Radiology, and Physical Medicine, University of Cordoba, 14004 Cordoba, Spain; 4Department of Morphological and Socio Sanitary Sciences, University of Cordoba, 14004 Cordoba, Spain

**Keywords:** Adhesive capsulitis, frozen shoulder, hydrodilatation, suprascapular nerve block

## Abstract

**Background**: Adhesive capsulitis (AC) causes a global limitation of both active and passive range of motion (ROM) in the shoulder, with or without pain, and no specific radiographic findings. Its course is self-limiting and progresses through three or four stages. The diagnosis is primarily clinical, since imaging tests are nonspecific. Treatment options include physical therapy (PT), intra-articular corticosteroid injections, suprascapular nerve block (SSNB), and hydrodilatation (HD). The latter is useful for expanding and reducing inflammation of the joint capsule through the insufflation of saline solution, anesthetics, and corticosteroids. **Objectives**: To compare whether patients with AC, stratified by phase 1 and 2, who receive high-volume HD as treatment achieve better outcomes in terms of shoulder pain and function compared to patients who receive low-volume HD. To compare whether there are differences in PT times and to determine mean axillary recess (AR) values. **Methods**: A randomized, parallel-block, triple-blind clinical trial will be conducted in 64 patients with AC in phases 1 and 2, aged 30 to 70 years, with limited active and passive ROM in two planes, and shoulder pain lasting more than 3 months. HD will be administered with volumes of 20 mL or 40 mL, followed by a conventional rehabilitation program. Outcomes will be reviewed at the 1st, 3rd, and 6th months of HD. Variables collected will include Shoulder Pain and Disability Index (SPADI), Visual Analog Scale (VAS), Range of motion (ROM), Lattinen index (LI), AR size, and time to completion of PT. **Results**: HD has been gaining clinical relevance in interventional rehabilitation as a treatment for AC, although its medium- and long-term efficacy remains a matter of debate. The variability in the volumes used for capsular expansion, with studies ranging from 18 mL to 47 mL, is compounded by the fact that most of these studies do not differentiate between AC stages. This could influence treatment effectiveness. Furthermore, diagnosis remains a challenge since valid and specific diagnostic parameters are lacking. **Conclusions**: Understanding the differences between HD techniques, considering the influence of certain factors such as the volume used or the stages of AC, as well as improving diagnosis and the coordination of scientific work. This could facilitate the development of protocols for the use of HD in AC.

## 1. Introduction

### 1.1. Background and Justification

Adhesive capsulitis (AC), also known as frozen shoulder, is defined as a condition in which a ”global limitation of active and passive shoulder movement, which may or may not be associated with pain, without other explanatory radiographic findings,” progressively develops [1]. The prevalence of this pathology is around 2–5% in the general population [2], with an age of onset between 40–60 years [3], and is up to four times more common in women [2,4].

Regarding the historical course of this disease, the clinical presentation was first described in 1872 by Duplay. Codman later named it “frozen shoulder” in 1934 [5]. Afterwards, Neviaser named it AC in 1945 [6], and he described its pathophysiological process.

This disease, with an average duration of 1 to 3 years [3], is a self-limiting process [4,5] that can be divided into three [2,4] or four [5,6] main phases according to the literature reviewed. This evolutionary process has been represented in Figure 1:

-**First Freezing Phase (0–9 months):** increasingly diffused and disabling pain, mostly at night, associated with mild stiffness. This phase includes the pre-freezing phase (0–3 months), with symptoms of mild pain and stiffness.-**Second Stiffness Phase (9–15 months):** Significant stiffness across all ranges of motion (ROM), accompanied by a progressive decrease in pain.-**Third Thawing Phase (15–24 months):** Gradual return of joint balance without associated pain.

Among the predisposing factors are gender due to estrogens [7], diabetes mellitus due to hyperglycemia [8], and thyroid diseases [9], all of them related to the creation of a chronic proinflammatory state that favors the appearance of AC. This chronic inflammation promotes the presence of inflammatory cells such as fibroblasts, B lymphocytes, cytokines, interleukins such as IL-6 or tumor necrosis factor alpha, which, from a histological point of view, favors the appearance of fibrosis, proliferative synovitis, and capsular thickening [2,4].

The pathological findings appear to be related to the phases of AC, with hypervascular and hypertrophic synovitis with fibroplasia and perivascular scarring being more common in phase 1; and hypercellular tissue accompanied by extensive fibroplasia but with a thin synovial membrane, without hypervascularity or synovitis, being more common in phase 2 [10].

At present, the general consensus regarding the diagnosis of AC is still to be developed due to its difficulty [11]. This has been based, so far, on fundamentally clinical aspects such as pain, and limitations of the joint ROM [3,4,5,12]. The limitation of both active and passive ROM is that the data are considered pathognomonic [2,4], with mainly a limitation of external rotation (ER) [3,4,5] and abduction (ABD) [12]. However, in a pathology in which the clinical stage in early phases seems to be associated with the need for shorter recovery times [13], as concluded in the systematic review by Schiltz M et al., its diagnosis should be based on more precise “diagnostic tools” in addition to the clinical pillar [12].

Currently, imaging tests focus on discarding pathologies that may simulate AC (rotator cuff tendinopathies, acromioclavicular osteoarthritis, labral injury, etc.) [2]. X-ray, despite being a basic and necessary test, is utilized to discard bone pathologies [14]. Magnetic resonance imaging (MRI) and ultrasound (US) appear to discard soft tissue pathologies.

Although US is not currently widely used for AC, there is evidence that it is as reliable as MRI in studying soft tissue parameters associated with AC [15,16]. “US can be incorporated into existing clinical diagnostic programs,” concluded a meta-analysis on the use of US in the diagnosis of AC published in 2020 [17].

Regarding possible treatments, there is one called benign negligence. This concept was previously used in the literature [13]. It refers to allowing the disease to progress given its self-limiting nature. However, other conservative treatments that may be useful in AC are now emerging. These include physical therapy (PT), intra-articular corticosteroid injections, suprascapular nerve block (SSNB), and hydrodilatation (HD) techniques [2,13,18,19,20].

Focusing on the latter, the procedure was first described in 1965 by Andren and Lundberg [21]. HD consists of an ultrasound-guided procedure in which a volume of fluid (saline, anesthetic, and corticosteroid) is injected to cause expansion of the joint capsule, which is already rigid [19,22].

Already in 2008, Cochrane concluded that HD was a useful tool in AC, but that it did not provide improvement in terms of pain and disability for more than 3 months [23]. The role that this technique plays in the medium to long term remains a topic of discussion due to the enormous “evidence gap” that exists [24].

### 1.2. Initial Hypothesis

Patients with AC, depending on the stage of progression, who receive high-volume HD as treatment, obtain better results on the Shoulder Pain and Disability Index (SPADI), the Visual Analog Scale (VAS), and ROM at the 1st, 3rd, and 6th months of therapy, compared to patients receiving low-volume HD in the general population.

### 1.3. Objectives

-To demonstrate the variability in the results of the main variables obtained between patients who received HD at different volumes.-To determine whether the results obtained differ when stratified by AC stage in phases 1 and 2.-To determine whether there are differences in the time to discharge from PT between patients who received HD at different volumes.-To determine the mean values that AR can present in AC.

### 1.4. Study Design

A triple-blind, parallel-block randomized clinical trial is proposed. It has been approved by ClinicalTrials.gov under the identification code: NCT06939530 (Protocol registered-version 2 on 23 April 2025.). It will be conducted in the Physical Medicine and Rehabilitation (PM&R) Department of the Hospital Universitario Reina Sofia of Cordoba (HURS), Spain.

This protocol has followed the quality standards based on the 2013 SPIRIT Declaration [25].

## 2. Materials and Methods

### 2.1. Participants—Selection Criteria

Patients are proposed to be selected from the PM&R service with a possible diagnosis of AC. The following selection criteria will be applied:Inclusion criteria: -Ages between 30 and 70 years.-Limited ROM, both active and passive, in two planes.-Shoulder pain that lasts more than 3 months.Exclusion criteria: -Lidocaine + trial with improved ROM [4]. Explanation in the Section 2.1.1 Clarifications regarding diagnosis and staging by phases.-Conditions that preclude treatment (active cancer, tissue infection, oral anticoagulant use, cardiac arrhythmias, etc.)-Previous HD treatment within the last year.-Stage 0 or 3 AC.-Non-adherence to the PT program, with non-attendance rates exceeding 20%.-Using X-rays and US to discard the presence of conditions that can cause symptoms similar to AC, such as acromioclavicular osteoarthritis, labral injury, full-thickness tear, massive rotator cuff tear, or rheumatic diseases.-Intra-articular injection of corticosteroids or SSNB in a period of less than 2 months.-Failed technique.

#### 2.1.1. Clarifications Regarding Diagnosis and Staging by Phases

In order to establish consistent diagnostic criteria and facilitate staging of AC, the selection process will be based on the following points.

Regarding diagnosis, the ROM limitation has to be affected in two planes, one of which must be the ER and/or ABD. These ROMs will be considered limited if they are reduced by equal to and/or more than 30% of what is considered normal or in relation to contralateral ROM. [3,22,26,27].

If there is any doubt as to whether the limitation may be due to a blockage imposed by the patient to prevent pain or not, a lidocaine test will be performed [3,4]. This consists of injecting 5–10 cc of 1% lidocaine into the subacromial space to facilitate passive examination of the joint. A positive result is found if ROM release and pain reduction are found after its use.

The phase staging will be performed according to the information provided in Section 1.1 Background and justification. Patients in stage 1 will be considered those whose predominant clinical manifestation is nocturnal pain accompanied by progressive limitation of ROM; and patients in stage 2, those in whom there is a limitation of ROM and a reduction in pain, in relation to the previous months. Patients with intermediate clinical manifestations, in whom both pain and limited ROM are present, will be classified according to the chronology of the disease, with the midpoint between the two stages being considered at 9 months of disease progression, as shown in Figure 1.

### 2.2. Sample Size [28,29,30,31]

The GRANMO calculator [28] was used for the sample calculation. Assuming an alpha risk of 0.05 and a statistical power greater than 0.8 in a bilateral contrast, 16 subjects in the 20 mL HD group and 16 in the 40 mL HD group are required to detect a difference equal to or greater than 17 units on the SPADI [31]. Since stratification is available, the baseline size for each phase is, therefore, n = 32 patients. The common standard deviation is assumed to be 15 points [29]. A 20% loss to follow-up rate has been estimated.

### 2.3. Interventions

#### 2.3.1. Type of Intervention

-HD technique with a 20 mL volume.-HD technique with a 40 mL volume.

#### 2.3.2. Procedures

It is proposed to perform HD techniques at different saline solution volumes. To achieve this:-Firstly, an ultrasound-guided SSNB will be performed with 4 mL of 0.25% bupivacaine + 0.5 mL of triamcinolone acetonide in the suprascapular notch. Figure 2.-Secondly, after 15 min of the SSNB, ultrasound-guided HD will begin. To perform this, the patient will be placed in a lateral decubitus position on the healthy arm. The arm to be treated will be positioned at the patient’s side without forcing its extension. The joint cavity will then be approached posteriorly, introducing a spinal needle in the ultrasound plane between the humeral cortex and the labrum. The joint cavity will then be confirmed by introducing physiological saline solution and verifying its reflux by pushing the plunger. Subsequently, 5 mL of 0.25% bupivacaine + 0.5 mL of triamcinolone acetonide will be introduced, topping up the predetermined volume with saline solution later. Figure 3.-Finally, while the anesthetic is still having effect and after HD, manipulations will be performed in both groups of patients using release techniques based on Kaltenborn mobilization and the soft tissue energy technique, performing it passively without forcing the pain, meaning that it will be stopped at the moment the patient feels pain [32].

#### 2.3.3. Failed Technique

The HD technique can sometimes be difficult and may not be performed as previously described. For this reason, there are criteria by which the technique is considered unsuccessful:-Acute pain that prevents the procedure from continuing at the patient’s request.-Inability to dilate the posterior joint capsule due to fluid leakage into soft tissue.-Collapse of the joint capsule.

#### 2.3.4. PT Protocol After HD

PT treatment will begin 3 to 5 days after HD. The patient will attend twice a week.Patients will be treated exclusively with manual techniques and kinesitherapy. They will be given manual techniques to release ROM and will be instructed on exercises adapted for both the gym and home development. Exercises that cause pain will be avoided.

Active PT: Four sets of 10 repetitions (rest between 30 s and 1 min between sets) twice a week: Self-passive pulley exercise: 1 min of activity, 1 min of rest for a total of 10 min.Active scapula exercises with anterior and posterior circular movements.Shoulder raises and lowers.Neck rotations.Neck tilt.Neck tilt and flexion stretch + tilt + rotations (chin to chest on one side and then the other). Hold for 20 s, then return to the starting position (three sets of three repetitions).Pole exercises: Flexion, extension, external and internal rotation, abduction (four sets of 10 repetitions).Biceps and triceps band exercises: diagonal curls inward and outward as tolerated, internal and external rotators, interscapularis, and lats (four sets of 10 repetitions).Manual therapy based on the Kaltenborn mobilization technique and the muscle energy technique [32].

### 2.4. Recruitment, Schedule, and Patient Flow (Figure 4)

**First Recruitment, patient assignment and interventional treatment phase** (Javier Muñoz Paz (J.M.-P)): The recruitment process will be conducted by J.M.-P during the initial consultation, who will select and classify patients referred to the PM&R service based on the criteria outlined above.The treatment assignment process will be pre-established and randomized. To this end, an Excel database has been created, where patients will be separated based on the AC phase of their disease, and each cell will be randomly assigned a treatment.While patients are included in the study, they will occupy the cells in the database. This assignment will only be known to J.M.-P, who will safeguard the information until the end of the study to avoid interference with the results obtained and, therefore, potential bias.In this initial consultation, the protocol will be explained to the patient, the specific informed consent form will be signed, and the variables outlined in Table 1 will be collected. J.M.-P will be responsible for these actions.Patients will then be scheduled for an interventional rehabilitation consultation. There, patients will undergo SSNB + HD, as explained in Section 2.3 Interventions, and they will be referred to the PT program through the mechanisms and means established for this purpose.**Second Phase of PT Treatment** (Francisco Espinosa Rueda (F.E.-R) and Amin Wahab Albañil (A.W.-A)): Referred patients will begin the PT program within 3 to 5 days. The PT start and discharge dates will be recorded according to pre-established criteria.**Third Phase of Review** (Ana Belén Jiménez Jiménez (A.B.J.-J)): Periodic reviews will be conducted at 1, 3, and 6 months after the intervention process to collect the corresponding variables. At the third month, A.B.J.-J will decide whether to repeat the HD and/or SSNB, according to the criteria established below: -Repeat HD: No 50% improvement in ROM was observed with respect to the total number of degrees that could be gained in relation to the initial situation and what is considered normal for each ROM, after 3 months of PT.-Repeat SSNB: A VAS score ≥ 7 precluding PT after 3 months of the previous SSNB.-MRI request [11,33]: If, after treatment, ROM does not improve 6 months after HD, according to the ROM recovery parameters for ABD, flexion (FLEX) ≥ 170°, and rotations ≥ 80°, described above, MRI will be considered to discard other pathologies. This information will be reflected in the report and in the final results of the study.**Fourth Phase of Statistical Analysis**: The collected information will be synthesized and entered into SPSS 24.0 software to obtain results. The following actions will be performed: ○For an independent data design: -Two groups: Student’s t-test or Mann-Whitney U-test.-More than two groups: Analysis of variance or Kruskal-Wallis H-test.○For a paired data design: -Two groups: Student’s t-test for paired data or Wilcoxon test.-More than two groups: Repeated measures analysis of variance or Friedman test.Before performing statistical tests, the validity of the analyses will be assessed. The normality of the variables will be evaluated using the Shapiro-Wilk test. The homogeneity of variances between groups will be analyzed using the Levene test. In the case of regression models, the linearity and independence of the residuals will be verified using scatter plots and residual analysis.A 2 × 4 mixed repeated measures ANOVA will be performed to examine the group × time interaction on the outcome variables, in order to determine whether the time course differs between the analyzed techniques.To correlate two quantitative variables, Pearson’s Linear Correlation Coefficient (r) will be used. For multiple comparisons, a test (Bonferroni, Finner, etc.) will be applied to correct the p-value. For multiple analysis, Multiple Linear Regression Analysis will be used.All contrasts will be two-tailed, and those with *p* < 0.05 will be considered significant. The data will be collected, processed, and analyzed using the statistical program SPSS v.24.

### 2.5. Blinding

The trial is designed as a triple-blind study. Only J.M.-P. will know the information regarding each patient’s treatment assignment. This information will be stored until the end of the study in the Excel database created by J.M.-P. Neither the patients, nor A.W.-A., nor A.B.J.-J.， nor F.E.-R., nor the statistician will have any knowledge of the treatment assignment.

### 2.6. Data Monitoring

To ensure proper follow-up of the patients included in the study, J.M.-P. will conduct monthly follow-up checks on the patients enrolled in the study.

The status of each of these patients will be updated, as well as the time and reason for withdrawal from the study. In addition, the research team works together and maintains daily communication.

All unforeseen situations that arise throughout this trial will be reflected in the medical reports and applications available at DIRAYA.

### 2.7. Data Collection and Management

The data obtained from the reviews conducted during the clinical trial will be stored in the DIRAYA system. From there, J.M.-P. will retrieve and enter them into a database, removing all sensitive information that could identify the patient.

This information will be entered into the SPSS 24.0 database and subsequently analyzed by an external statistician. The statistician will also be unaware of sensitive information or the treatment assignment of each patient.

## 3. Results

### Variables/Sources of Information

SPADI [22,29,31,34].

SPADI is an outcome measure widely used in studies of rotator cuff pathology. It provides information about pain and the limitation of shoulder pathologies.

○Pain Scale

How severe is the pain? 0 = no pain and 10 = the worst pain imaginable.

-At its worst?-When lying on the involved side?-When reaching for something on a high shelf?-When touching the back of your neck?-When pushing with the affected arm?

○Disability Scale:

How much difficulty do you have? 0 = no pain and 10 = the worst pain imaginable.

-Washing your hair?-Washing your back?-Putting on a T-shirt or sweater?-Putting on a shirt that buttons down the front?-Putting on your pants?-Placing an object on a high shelf?-Carrying a heavy object of 10 pounds (4.5 kg)-Taking something from your back pocket?

Pain and disability are calculated separately, and a percentage of impairment is obtained jointly.

It has been shown that the minimal clinically important difference (MCID) for changes in SPADI should be ≥17 points.

This index has demonstrated “good internal consistency, convergent validity, and reliability” in its Spanish version, making it one of the most reliable measurement instruments in the field of shoulder disorders. It is measured based on percentages.

VAS [14,35,36].

The VAS “is a validated subjective measure for acute and chronic pain.” It allows researchers to measure the pain intensity with maximum reproducibility.

It consists of a 10 cm horizontal line, at the ends of which are the extreme expressions of a symptom. On the left (0 cm) is the absence or least intensity, and on the right (10 cm) is the greatest intensity. The patient is asked to mark the point on the line that indicates the intensity. It is measured using points. The MCID for the symptom state to be acceptable is usually two points.

ROM [21,37].

ROM assessment, measured in degrees (°), is a basic practice in the study of shoulder pathologies, especially in the case of AC. ROM should be measured both actively and passively.

The ROMs that will be assessed actively and passively will primarily be FLEX, ABD, ER with the arm at 90° of abduction, asking the patient to show us the palm of their hand, and internal rotation (IR) with the arm at 90° of abduction, asking the patient to show us the back of their hand. All of these are measured with the PLURIMETER inclinometer [38,39].

The measurement will be performed with the patient actively seated, and if limitations are found, the patient will be placed supine to eliminate the influence of gravity.

LATTINEN INDEX (LI) [40].

The LI is a widely used tool for pain assessment, validated in Spain as a tool to measure the degree of pain in patients with chronic pain. This scale consists of five items scored from 0 to 4:Pain intensity.Pain frequency.Analgesic use.Degree of disability.Hours of sleep.

Although this test is widely used in chronic pain conditions, its use in AC is not very common. However, we believe it can provide important data such as analgesic medication intake. This is why we consider its use appropriate.

Their values are also generally compatible with VAS assessments.

Correlation of VAS and LI levels [40]VAS 0–2.5 → IL = 6 VAS 2.5–5 → IL = 10 VAS 5–7.5 → IL 13 VAS 7.5–10 → IL = 14

PATIENT GLOBAL IMPRESSION OF IMPROVEMENT SCALE (PGI-I) [41]

The PGI-I consists of a single question asking the patient to rate the change from their baseline situation after treatment, among the following items:

1. Very much better 2. Much better 3. A little better 4. No change 5. A little worse 6. Much worse 7. Very much worse.

Scores of “Very much better” or “Much better” are generally considered successful.

Although this scale is not commonly used in AC, this research team believes that its use can provide useful information regarding the patient’s pre-treatment expectations.

CLINICAL GLOBAL IMPRESSION OF GLOBAL IMPROVEMENT SCALE (CGI-GI) [41]

The CGI-GI consists of a single question in which the healthcare professional must classify the patient’s change from baseline after treatment, among the following items:

1. Very much better 2. Much better 3. A little better 4. No change 5. A little worse 6. Much worse 7. Very much worse.

Although this scale is not commonly used in AC, this research team believes that its use can provide useful information regarding the comparison of the healthcare professional’s pre-treatment expectations.

AXILLARY RECESS (AR) SIZE [15,42].

Measured using US with a longitudinal section of the AR in millimeters (mm). The patient will be placed supine with the shoulder abducted at 90° and the elbow flexed. The AR measurement will be performed only by J.M.-P, during the initial consultation.

TIME FROM START TO END OF PT.The following criteria will be established for discontinuing PT treatment:○Recovery of ROM for ABD, FLEX ≥ 170° and ≥80° in rotations.○If a repeated HD or SSNB is necessary after the third month, the time will continue to be counted until either: -Recovery of ROM for ABD, FLEX ≥ 170° and ≥80° in rotations.-Pain reduction with VAS ≤ 2.-Six-month follow-up after a new procedure.REPEATING A NEW HD.The following criteria will be established in order to perform a new HD:○No 50% improvement in ROM was observed with respect to the total number of degrees that could be gained in relation to the initial situation and what is considered normal for each ROM, after 3 months of PT.REPEATING A NEW SSNB.The criteria for performing a new SSNB will be:○A VAS score of ≥7 precluding PT after 3 months of the previous SSNB.

## 4. Discussion

Most of the studies reviewed for this study show a heterogeneous pattern regarding the results of this technique and the comparison with intra-articular injections. Some of them conclude that HD has an “efficacy comparable to intra-articular corticosteroid injections” [43]. However, studies are beginning to show the long-term benefits it can provide, such as the study by Sofía Dimitri—Pinheiro et al., which concludes that “it is effective in improving shoulder pain and increasing disability with benefits for a period of up to 2 years” [44].

The difficulty in finding consistent results using this technique may arise from the lack of a standardized approach. This is why this essay focuses on various points of conflict.

The first of these differences concerns the volume required to achieve optimal capsular expansion. Reviewing the literature, we found that the volumes used in different studies vary greatly. In 2020, Jang Hyuk Cho concluded that the volume required to achieve optimal expansion was around 18 mL [45]. This same volume parameter, or similar values, also appeared in later studies on HD [14,46]. However, more up-to-date studies began to report HD with higher volumes, typically 30–40 mL [47]. In 2023, Sofia Dimitri-Pinheiro reported that her study used volumes close to 50 mL [44], as did the study by Magdalena Pimenta et al., which reported that a volume of up to 47 mL [48] allowed an optimal capsular expansion. These last two studies and some others agree that volumes greater than the initial 18 mL did not pose a risk of capsular rupture, a finding associated with worse outcomes [44,45,48].

This difference in volumes is the main reason for deciding to compare 20 mL vs. 40 mL in HD in this clinical trial.

The combined use, or not, of other therapies with HD, such as SSNB and PT, is also a topic of interest. However, their use is widely common and seems not to be receiving enough discussion. In the case of SSNB, its use has been associated with additive improvements in both pain and shoulder function [18,49], being in many cases “a valuable complementary therapy” [50]. In the case of PT, its use in the first line of treatment [20] is more than established in clinical practice. Most studies show that PT is a very useful complementary treatment [14,18,47,51,52,53].

The second important point, not because of the scientific discussion, but rather due to the absence of this variable in the reviewed studies, is the lack of stratification by phases of this disease. If we take into account the existing relationship between the clinical phases and the pathological findings, the fact that studies on HD are reviewed from 2019 to the present and the majority do not differentiate these results according to the phase of AC [18,22,43,44,46,47,48] does not make much sense. Only in the study by Fabio Vita et al. in 2024 is a differentiation made between the results obtained in both phase 1 and phase 2 [14].

Diagnosis remains a topic of great debate. Regarding MRI, although there is still no consensus, this test provides parameters that may be useful in cases of contradictory examination [54]. However, if we consider the long waiting lists, the need for several appointments to associate clinical symptoms with images and, perhaps, the most devastating data associated with the study carried out by Dimitris MD et al. in which it is stated that “the routine use of shoulder MRI scans in patients with AC but without suspicion of an additional pathology may not be indicated”, because “37 MRIs were necessary to change a single therapeutic plan” [33] makes us think that MRI may not be the diagnostic tool we are looking for.

This idea agrees with the protocol proposed by Riccardo Picasso et al. in 2023, which stated that “MRI should be considered in doubtful cases in order to reveal differential factors” [11] with diseases with similar clinical features such as labral involvement.

The opposite occurs with the use of US. The possibility of performing it at the same time as the consultation, the high availability for many specialties, the low cost, and the good diagnostic accuracy [11] make this test a great possibility of diagnostic support in AC. Within the ultrasound parameters related to AC, the measurement of the thickness of the AR is a piece of data that, as shown in the study by Byung Chan L et al., seems to correlate significantly with pain, functionality, and ROM [42]. The average scores that are taken as a reference to determine whether AC may exist would be, together with clinical data, an AR of ≥4 mm, although more studies are needed to demonstrate this [11,42,55].

If we take into account everything previously stated, the variability between HD techniques, the lack of phase stratification in the results, the diagnostic difficulty and, above all, the absence of “high quality scientific work” [56], it is understandable that the most repeated conclusion is the need to carry out more studies in order to protocolize this treatment [18,20,43,47].

## 5. Conclusions

The use of HD in clinical rehabilitation practice as a second-line treatment for AC is increasingly common. Despite this, its protocolization is still far from a reality.

This clinical trial would facilitate obtaining consistent results and, therefore, more robust clinical recommendations by addressing two potentially decisive factors in the results obtained with the use of HD: the volume used to achieve effective capsular expansion and the lack of stratification according to clinical stages of AC.

Furthermore, it could provide more information in an area yet to be developed: diagnosis. Specific parameters need to be identified to complement conventional physical examination. In this regard, measuring the AR of the axillary recess appears to be a promising option.

Therefore, this trial could facilitate the understanding between the technical differences in HD, taking into account the influence of factors such as the volume used or the stages of AC, in addition to improving diagnosis and the coordination of scientific work. This could ultimately be reflected in the development of protocols for the use of HD in AC.

## Figures and Tables

**Figure 1 clinpract-15-00141-f001:**
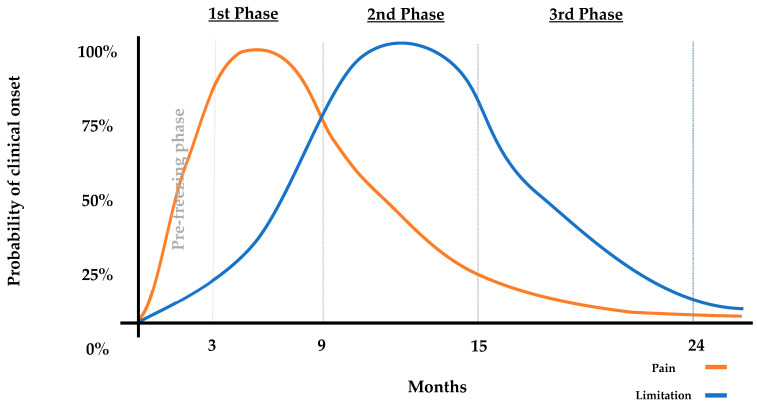
**Graphical representation of the evolutionary process of AC**. The graph divides AC into three phases (first phase includes the pre-freezing phase). Throughout these phases, both pain and ROM limitation appear. The orange line (--) represents the pain evolutive process and the blue line (--) represents limitation evolutive process.

**Figure 2 clinpract-15-00141-f002:**
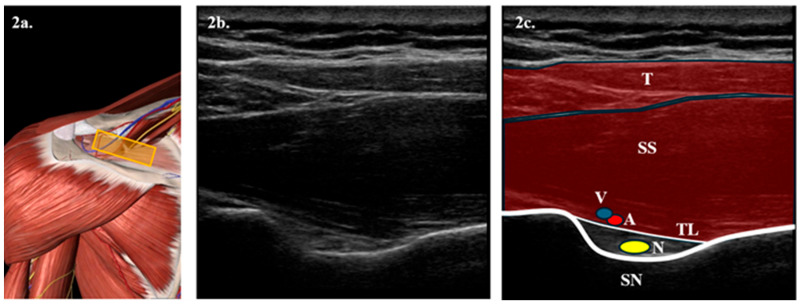
**Illustration of access point for performing SSNB via posterior route:** (**a**) Anatomical representation of the scapular fossa seen from a posterior view, obtained from Essential Anatomy 5 after removing the trapezius muscle plane. The yellow rectangle represents the probe position. (**b**) Longitudinal ultrasound section with a 6–15 MHz linear probe of the suprascapular fossa. (**c**) Representation of the most notable anatomical structures in the suprascapular fossa. The foreground corresponds to the trapezius (T), followed by the supraspinatus (SS) muscle. Below, the transverse ligament (TL), which forms the roof of the scapular notch (SN), appears. Below the TL, the suprascapular nerve (N), represented by a yellow circle, appears; and above, the suprascapular vein (V) and artery (A) appear represented by a blue and red circle.

**Figure 3 clinpract-15-00141-f003:**
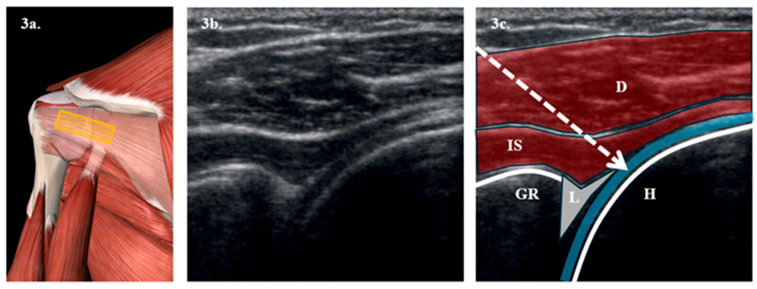
**Illustration of access point for posterior approach HD technique**. (**a**) Anatomical representation of the posterior articular recess seen from behind, obtained from Essential Anatomy 5 after removing the plane of the deltoid muscle. The yellow rectangle represents the position of the probe. (**b**) Longitudinal ultrasound section with a 6–15 MHz linear probe of the posterior articular recess. (**c**) Representation of the most notable anatomical structures in the posterior articular recess. In the foreground, we see the deltoid muscle (D) and below it, the infraspinatus muscle (IS). Two hyperechoic lines are observed, corresponding to the cortices of the glenohumeral recess (GR) and the humeral head (H). Between them, the labrum (L) appears, represented as a gray triangle. The dashed arrow represents the path of the needle toward the joint cavity, represented in blue.

**Figure 4 clinpract-15-00141-f004:**
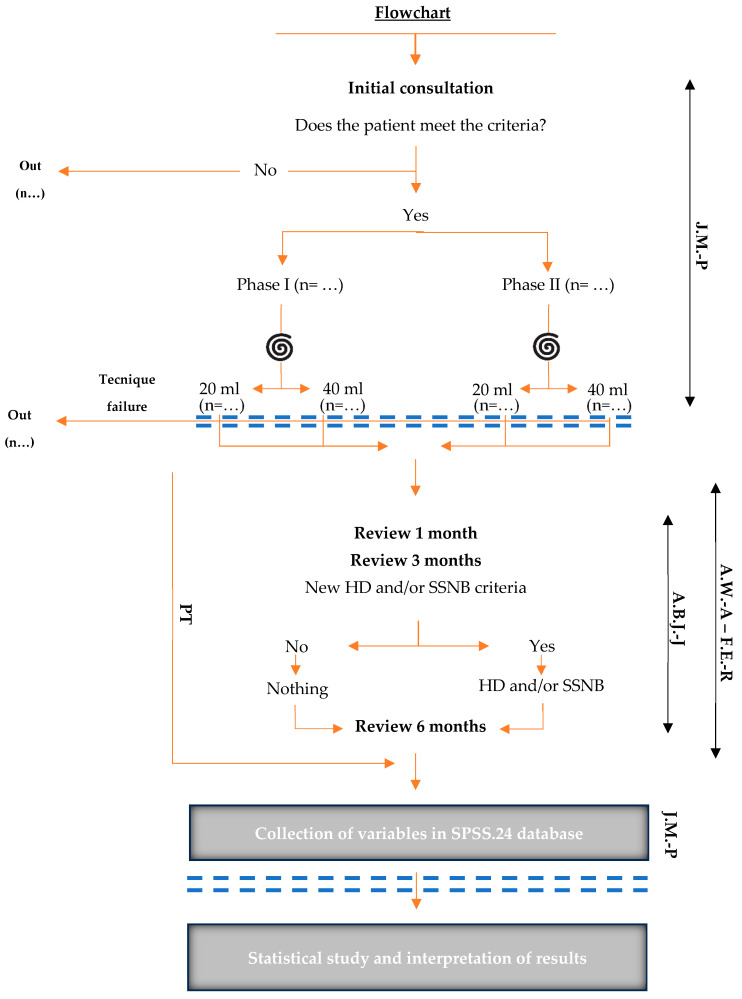
**Patient flowchart throughout the clinical trial.** The orange arrows indicate the transition of patients from one phase to another. The spiral (
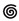
) represents the time of randomization of each patient, and the blue dashed lines (**= =**) represent the blinding times of the study. Each phase is assigned a collaborator, represented by the black arrow(↔) on the right side of the flowchart: Recruitment, interventional treatment phase and collection of variables—Javier Muñoz Paz (J.M.-P.); physical therapy (PT) treatment phase with Francisco Espinosa Rueda (F.E.-R.) and Amin Wahab Albañil (A.W.-A.); and review phase—Ana Belén Jiménez Jiménez (A.B.J.-J.).

**Table 1 clinpract-15-00141-t001:** Chronology of variables to be collected according to consultation. The × represents the exact moment where the variables will be taken. ***** Explanation of the clinical trial, screening, and signing of informed consent. The ↔ represents the period from when the PT starts until it ends at the latest, and it may end earlier if the pre-established criteria are met.

Variables	Initial Consultation *	Intervention	1 Month	3 Months	6 Months
**Age**	×				
**Gender**	×				
**Previous** **treatments**	×				
**Diseases**	×				
**Time since onset** **of symptoms**	×				
**AR size**	×		×	×	×
**SPADI**	×		×	×	×
**VAS**	×		×	×	×
**ROM**	×		×	×	×
**IL**			×	×	×
**PGI-I**					×
**CGI-CI**					×
**Duration of the PT**			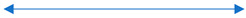
**Repeat HD**				×	
**Repeat SSNB**				×	
**MRI request**				×	

## Data Availability

The database obtained for the analysis of this study will be made available to those researchers who request it. When the data are analyzed, they will be published in full detail in open access, except for those that affect the confidentiality of the participants. On this database, any type of personal information that allows patients to be identified will be eliminated.

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
