# Peer review of "Effects of Hydrodilatation at Different Volumes on Adhesive Capsulitis in Phases 1 and 2: Clinical Trial Protocol HYCAFVOL"

_clinpract, 2025, doi:10.3390/clinpract15080141_

Round 1

Reviewer 1 Report

Comments and Suggestions for Authors

The study is well arranged and the protocol clear

Maybe the Authors need to emprove the Intro and discussion adding best references 

Author Response

First of all, we would like to sincerely thank you for the time and effort dedicated to reviewing our manuscript. We truly appreciate your positive comment noting that our study is well organized and that the protocol is clearly presented.

Regarding your observation: “Maybe the Authors need to improve the Intro and Discussion adding best references”, we completely agree with your suggestion. We believe that enhancing these sections with more relevant and high-quality references significantly strengthens the manuscript.

We leave you here the main changes that have been made in the new text:

  • The paragraph located between lines 90-93 has been modified. It has been rewritten and reinforced with four articles in lines 95 (14), 98 (15, 16) and line 100 (17).

14. Vita, F.; Pederiva, D.; Tedeschi, R.; Spinnato, P.; Origlio, F.; Faldini, C. Adhesive capsulitis: the importance of early diagnosis and treatment. J Ultrasound. 2024, 27(3), 579–87.

15. Stella, S.M.; Gualtierotti, R.; Ciampi, B.; Trentanni, C.; Sconfienza, L.M.; Del Chiaro, A. Ultrasound features of adhesive capsulitis. Rheumatol Ther. 2022, 9(2), 481–95.

16. Al Khayyat, S.G.; Falsetti, P.; Conticini, E.; Frediani, B.; Galletti, S.; Stella, S.M. Adhesive capsulitis and ultrasound diagnosis, an inseparable pair: a novel review. J. Ultrasound 2022, 26(2), 369–384.

17. Wu, H.; Tian, H.; Dong, F.; Liang, W.; Song, D.; Zeng, J.; et al. The role of grey-scale ultrasound in the diagnosis of adhesive capsulitis of the shoulder: a systematic review and meta-analysis. Med. Ultrason. 2020, 22(3), 305.

  • The exclusion criteria has been developed in lines 152-156, in order to include imaging tests.

  • The patients classification methodology is explained in lines 175-181.

  • A new reference has been added in order to reinforce the use of the SPADI scale in the calculation of the sample size. Line 185 (30).

30. Parashar, A.; Goni, V.; Neradi, D.; Guled, U.; Rangasamy, K.; Batra, Y.K. Comparing three modalities of treatment for frozen shoulder: a prospective, double-blinded, randomized control trial. Indian J. Orthop. 2021, 55(2), 449–456.

  • The HD final process has been specified in lines 213-218.

  • A new criteria has been added for the use of the MRI at the sixth month after the HD. Lines 283-286 with their corresponding references (11, 32).

11. Picasso, R.; Pistoia, F.; Zaottini, F.; Marcenaro, G.; Miguel-Pérez, M.; Tagliafico, A.S. Adhesive Capsulitis of the Shoulder: Current Concepts on the Diagnostic Work-Up and Evidence-Based Protocol for Radiological Evaluation. Diagnostics. 2023, 13(22), 3410.

32. Dimitriou, D.; Winkler, E.; Zindel, C.; Grubhofer, F.; Wieser, K.; Bouaicha, S. Is routine magnetic resonance imaging necessary in patients with clinically diagnosed frozen shoulder? Utility of magnetic resonance im-aging in frozen shoulder. JSES Int. 2022, 6(5), 855–8.

  • More information related to the statistical analysis mechanism has been added in lines 296-300.

  • In lines 425-426, we have included that Dr. Javier Muñoz Will be in charge of doing the measurements of the axillary recess

  • The phrase “full recovery of ROM after 4 sessions” has been changed for “Recovery of ROM for ABD, FLEX ≥170° and ≥80° in rotations.”. Lines 430-431 y 434.

  • The paragraph related to the volume comparison in the discussion, between lines 462-482, has been modified, ending up like this:

The first of these differences concerns the volume required to achieve optimal capsular expansion. Reviewing the literature, we found that the volumes used in different studies vary greatly. In 2020, Jang Hyuk Cho concluded that the volume required to achieve optimal expansion was around 18 ml [44]. This same volume parameter, or similar values, also appeared in later studies on HD [14,45]. However, more up-to-date studies began to report HD with higher volumes, typically 30–40 ml [46]. In 2023, Sofia Dimitri-Pinheiro reported that her study used volumes close to 50 ml [43], as did the study by Magdalena Pimenta et al., which reported that a volume of up to 47 ml [47] allowed an optimal capsular expansion. These last two studies and some others agree that volumes greater than the initial 18 ml did not pose a risk of capsular rupture, a finding associated with worse outcomes [43,44,47].

This difference in volumes is the main reason for deciding to compare 20 ml vs. 40 ml in HD in this clinical trial.”

14. Vita, F.; Pederiva, D.; Tedeschi, R.; Spinnato, P.; Origlio, F.; Faldini, C. Adhesive capsulitis: the importance of early diagnosis and treatment. J Ultrasound. 2024, 27(3), 579–87.

43. Dimitri-Pinheiro, S.; Klontzas, M.E.; Vassalou, E.E.; Pimenta, M.; Soares, R.; Karantanas, A.H. Long-term outcomes of ultrasound-guided hydrodistension for adhesive capsulitis: a prospective observational study. Tomography. 2023, 9(5), 1857–67.

44. Cho, J.H. Updates on the treatment of adhesive capsulitis with hydraulic distension. Yeungnam Univ J Med. 2021, 38(1), 19–26.

45. Swaroop, S.; Gupta, P.; Patnaik, S.; Reddy, S. Intra-articular steroid alone vs hydrodilatation with intraarticular steroid in frozen shoulder – a randomised control trial. Malays Orthop J. 2023, 17(1), 34–9.

46. Whelan, G.; Yeowell, G.; Littlewood, C. Patient experiences of hydrodistension as a treatment for frozen shoulder: a longitudinal qualitative study. PLoS One. 2024, 19(6), e0304236.

47. Pimenta, M.; Vassalou, E.E.; Klontzas, M.E.; Dimitri-Pinheiro, S.; Ramos, I.; Karantanas, A.H. Ultrasound-guided hydrodilatation for adhesive capsulitis: capsule-preserving versus capsule-rupturing technique. Skeletal Radiol. 2024, 53(2), 253–61.

  • Five new references in favor of using FST have been added in line 489.

14. Vita, F.; Pederiva, D.; Tedeschi, R.; Spinnato, P.; Origlio, F.; Faldini, C. Adhesive capsulitis: the importance of early diagnosis and treatment. J Ultrasound. 2024, 27(3), 579–87.

18. Hill, J.L. Evidence for Combining Conservative Treatments for AdhesiveCapsulitis. Ochsner Journal. 2024, 24(1), 47–52.

46. Whelan, G.; Yeowell, G.; Littlewood, C. Patient experiences of hydrodistension as a treatment for frozen shoulder: a longitudinal qualitative study. PLoS One. 2024, 19(6), e0304236.

50. Cho, C.H.; Bae, K.C.; Kim, D.H. Treatment strategy for frozen shoulder. Clin Orthop Surg. 2019, 11(3), 249.

51. Di Mascio, L.; Hamborg, T.; Mihaylova, B.; Kassam, J.; Shah, B.; Stuart, B.; et al. The Adhesive Capsulitis Cortico-steroid and Dilation (ACCorD) randomized controlled trial. Bone Jt. Open 2024, 5(3), 162–173.

52. Kayaokay, K.; Arslan Yurtlu, D. A comparison of the treatment outcomes with and without the use of intra-articular corticosteroids for frozen shoulder manipulation. Cureus 2023, 15(8), e44427.

  • It has been included a new paragraph which explains the need of using MRI in cases presenting diagnostic difficulty in lines 506-508, with its corresponding reference (11).

11. Picasso, R.; Pistoia, F.; Zaottini, F.; Marcenaro, G.; Miguel-Pérez, M.; Tagliafico, A.S. Adhesive Capsulitis of the Shoulder: Current Concepts on the Diagnostic Work-Up and Evidence-Based Protocol for Radiological Evaluation. Diagnostics. 2023, 13(22), 3410.

We hope these revisions appropriately address your recommendation and contribute to enhancing the scientific quality of the manuscript.

Once again, thank you for your valuable comments.

Reviewer 2 Report

Comments and Suggestions for Authors

Thank you for your submission on a very interesting technique. Please clarify how the volumes were determined. It would seem as if a certain level of increased intra-capsular pressure is required along with a certain volume of fluid to achieve capsular expansion. Therefore, in the failed treatment cases, how can you be certain that enough fluid was insufflated to stretch the capsule? Next, how long was the fluid retained in the shoulder joint? If longer than a few days, how did you verify that the fluid and capsular stretch persisted rather than the fluid being resorbed into the test subject's body ( which happens after shoulder arthroscopy)? Overall, the methods section needs clarification.

Author Response

We would like to sincerely thank the reviewer for the time and effort invested in evaluating our manuscript and for the insightful comments that help improve the clarity and scientific rigor of our work. Below, we address each of the points raised.

Comment 1:

Please clarify how the volumes were determined. It would seem as if a certain level of increased intra-capsular pressure is required along with a certain volume of fluid to achieve capsular expansion.

Response 1:

Estimated reviewer, with the objective of solving doubts about the raised question, there has been a modification of the paragraph regarding the volume comparison in the discussion in lines 462-482. I tended up like this:

The first of these differences concerns the volume required to achieve optimal capsular expansion. Reviewing the literature, we found that the volumes used in different studies vary greatly. In 2020, Jang Hyuk Cho concluded that the volume required to achieve optimal expansion was around 18 ml [44]. This same volume parameter, or similar values, also appeared in later studies on HD [14,45]. However, more up to date studies began to report HD with higher volumes, typically 30–40 ml [46]. In 2023, Sofia Dimitri-Pinheiro reported that her study used volumes close to 50 ml [43], as did the study by Magdalena Pimenta et al., which reported that a volume of up to 47 ml [47] allowed an optimal capsular expansion. These last two studies and some others agree that volumes greater than the initial 18 ml did not pose a risk of capsular rupture, a finding associated with worse outcomes [43,44,47].

This difference in volumes is the main reason for deciding to compare 20 ml vs. 40 ml in HD in this clinical trial.”

14. Vita, F.; Pederiva, D.; Tedeschi, R.; Spinnato, P.; Origlio, F.; Faldini, C. Adhesive capsulitis: the importance of early diagnosis and treatment. J Ultrasound. 2024, 27(3), 579–87.

43. Dimitri-Pinheiro, S.; Klontzas, M.E.; Vassalou, E.E.; Pimenta, M.; Soares, R.; Karantanas, A.H. Long-term outcomes of ultrasound-guided hydrodistension for adhesive capsulitis: a prospective observational study. Tomography. 2023, 9(5), 1857–67.

44. Cho, J.H. Updates on the treatment of adhesive capsulitis with hydraulic distension. Yeungnam Univ J Med. 2021, 38(1), 19–26.

45. Swaroop, S.; Gupta, P.; Patnaik, S.; Reddy, S. Intra-articular steroid alone vs hydrodilatation with intraarticular steroid in frozen shoulder – a randomised control trial. Malays Orthop J. 2023, 17(1), 34–9.

46. Whelan, G.; Yeowell, G.; Littlewood, C. Patient experiences of hydrodistension as a treatment for frozen shoulder: a longitudinal qualitative study. PLoS One. 2024, 19(6), e0304236.

47. Pimenta, M.; Vassalou, E.E.; Klontzas, M.E.; Dimitri-Pinheiro, S.; Ramos, I.; Karantanas, A.H. Ultrasound-guided hydrodilatation for adhesive capsulitis: capsule-preserving versus capsule-rupturing technique. Skeletal Radiol. 2024, 53(2), 253–61.

Comment 2:

Therefore, in the failed treatment cases, how can you be certain that enough fluid was insufflated to stretch the capsule?

Response 2:

Estimated reviewer, regarding the case raised it is explained in “The failed technique” section in line 220. It an exclusion criteria which will be reflected in the descriptive results. This decision was established with the aim of being faithful to the random assignment of treatment

Our study does not specify the reasons for a leak, since loss of joint fluid would not make HD optimal, and therefore could be a confounding factor. In short, these patients would be excluded.

I'm leaving the section on "Failed Technique" here, hoping you enjoy what I've explained.

Failed technique.

The HD technique can sometimes be difficult and may not be performed as previously described. For this reason, there are criteria by which the technique is considered unsuccessful:

- Acute pain that prevents the procedure from continuing at the patient's request.

- Inability to dilate the posterior joint capsule due to fluid leakage into soft tissue.

- Collapse of the joint capsule.

Comment 3:

Next, how long was the fluid retained in the shoulder joint? If longer than a few days, how did you verify that the fluid and capsular stretch persisted rather than the fluid being resorbed into the test subject's body (which happens after shoulder arthroscopy)?

Response 3:

Estimated reviewer, this question is not tackled in our research. In our case, we take as optimal, a capsular dilation seen by ultrasound with the volume determined for each participant, in addition to not complying with what is stated in the "Failed technique"

Comment 4:

Overall, the methods section needs clarification.

Response 4:

We appreciate this feedback and have attempted to improve this section with the following changes.

  • We will add to the inclusion criteria the need for imaging tests that were not previously reflected in lines 152 and 156:

Using X-rays and US, to discard the presence of conditions that can cause symptoms similar to AC, such as acromioclavicular osteoarthritis, labral injury, massive rotator cuff tear, or rheumatic diseases.”

  • Staging will be based on the patient's clinical evaluation, as described in lines 61–68. In this case, no cutoff points have been established for VAS or ROM. To clarify this staging, we have rewritten lines 175–181 as follows:

Patients in stage 1 will be considered those whose predominant clinical manifestation is nocturnal pain accompanied by progressive limitation of ROM; and patients in stage 2, those in whom there is a limitation of ROM and a reduction in pain, in relation to the previous months.”

  • We have modified the sample size section on lines 186 – 192 and added a new reference:

Assuming an alpha risk of 0.05 and a statistical power greater than 0.8 in a bilateral contrast, 16 subjects in the 20 ml hydrodilation group and 16 in the 40 ml hydrodilation group are required to detect a difference equal to or greater than 15 units on the SPADI. Since stratification is available, the baseline size for each phase is therefore n=32 patients ”.

30. Parashar, A.; Goni, V.; Neradi, D.; Guled, U.; Rangasamy, K.; Batra, Y.K. Comparing three modalities of treatment for frozen shoulder: a prospective, double-blinded, randomized control trial. Indian J. Orthop. 2021, 55(2), 449–456.

  • We have clarified the information on lines 213 – 220 regarding handling after HD, which is now as follows:

Finally, while the anesthetic is still having effect and after HD, manipulations will be performed in both groups of patients using release techniques based on Kaltenborn mobilization and the soft tissue energy technique, performing it passively without forcing the pain, meaning that it will be stopped at the moment the patient feels pain.”

  • It has been added in line 267 that in the phase of “1st Recruitment, patient assignment and interventional treatment phase”, “J.M.-P will be responsible for these actions.”

  • A new paragraph has been added regarding the requirements for performing an MRI between lines 283-286, with the following information:

MRI request (11, 32): If, after treatment, ROM does not improve 6 months after HD, according to the ROM recovery parameters for abduction, flexion ≥170°, and rotation ≥80°, described above, a MRI will be considered to discard other pathologies. This information will be reflected in the report and in the final results of the study.”

11. Picasso, R.; Pistoia, F.; Zaottini, F.; Marcenaro, G.; Miguel-Pérez, M.; Tagliafico, A.S. Adhesive Capsulitis of the Shoulder: Current Concepts on the Diagnostic Work-Up and Evidence-Based Protocol for Radiological Evaluation. Diagnostics. 2023, 13(22), 3410.

32. Dimitriou, D.; Winkler, E.; Zindel, C.; Grubhofer, F.; Wieser, K.; Bouaicha, S. Is routine magnetic resonance imaging necessary in patients with clinically diagnosed frozen shoulder? Utility of magnetic resonance imaging in frozen shoulder. JSES Int. 2022, 6(5), 855–8.

  • New information has been added regarding the statistical analysis between lines 296 – 300:

Before performing statistical tests, the validity of the analyses will be assessed. The normality of the variables will be evaluated using the Shapiro-Wilk test. The homogeneity of variances between groups will be analyzed using the Levene test. In the case of regression models, the linearity and independence of the residuals will be verified using scatter plots and residual analysis.”

All modifications are marked in the revised manuscript and summarized in the tracked changes document for ease of review.

We are grateful once again for your thorough review and constructive comments, which have helped us improve the clarity and transparency of our study. We remain at your disposal for any further revisions or elaborations you may require.

Reviewer 3 Report

Comments and Suggestions for Authors

This is a very interesting, proposed trial. Regarding the current knowledge about adhesive capsulitis is very worthwhile and the results should be very helpful in the management of patients with capsulitis. The use of the word idiopathic will further clarify the study.

The introduction was done well. I would agree with line 81 that the diagnosis of capsulitis is still to be developed. I would disagree with lines 89-91. There is no mention here of the value of a plain x-ray or that glenohumeral joint arthritis may resemble capsulitis. There is only one reference cited for this statement which was rather obscure with no authors. I consider that for a patient presenting with a painful stiff shoulder that a plain x-ray is still a good baseline investigation. I do not see that ultrasound really has much place, and there is no reference cited to support this. In line 95 it is stated that physical therapy may be useful in the management of capsulitis. To my knowledge there is no good study that has ever demonstrated this. Again Reference 2 which has no authors was cited.

Concerning the materials and methods:

In the inclusion criteria there is no mention of any imaging. It should be mentioned in the inclusion criteria that there should be no evidence of any osteoarthritis on any imaging. I consider a X-Ray should always be obtained in this situation.  As mentioned later in the paper it is probably not cost effective to obtain an MRI scan in all cases. However, it could possibly be stated in the inclusion criteria that if there is any doubt as to the status of the glenohumeral joint on plain radiographs that an MRI scan would be obtained. Line 141 exclusion : non- adherence to physiotherapy programme – is  this selection bias as little good literature to support physiotherapy as a successful treatment for  AC.

 There is also no exclusion of patients with a clinically proven full thickness cuff tear.

It is stated that differentiation as to whether pain or stiffness is a predominant feature when assigning the patients to stage 1 or 2. Is this based on the patient physical assessment or on the recorded VAS and ROM?

2.3.2. Procedures:

In line 190 it is stated that there is topping up the predetermined volume with saline solution later. Figure 3 is quoted, however the illustration for this appears to be figure 2. So I gather from figure 4 that the predetermined volume of saline solution was either 20ml or 40ml. In our department the hydrodilatations are done under fluoroscopy. We have found that in a number of cases of severe capsulitis that it is not possible to inject 40ml into the glenohumeral joint. Also some capsules actually rupture through the subscapularis bursa before 40mls has been injected. It needs to be stated how these cases with a very contracted capsule, which allow less than 40ml, would be dealt with under the protocol.

It is then stated in line 191 that there is then a release manipulation. I gather that this is carried out while the local anaesthetic is still within the joint, but this needs to be stated.Is this for both 20 and 40 Mls. I note the comments regarding the failed technique. I note in line 199 that one of these is inability to dilate the capsule due to fluid leakage into the soft tissue. This could be related not to a full thickness tearing of the rotator cuff, but a poor rotator cuff  which inadequately maintains the fluid in the glenohumeral joint and allows it to leak into the subacromial bursa. Would these cases be still included under the current study protocol, or would, as stated, the technique then be considered unsuccessful and these patients be discarded from the study?

Line 292, who is diagnosing phase 1 and Phase 2 and assigning the patient.

Line 346. Is the Lattinen Index for chronic pain an appropriate for assessment for patients’ pain level in AC  given the progression and relatively short time range of phase 1 ?

In lines 363 and 375 there are two improvement scales utilised. I found it difficult to determine the difference between these two scales as both of them involved a single question, and the difference between the two needs to be better explained. Neither seems to address or specify pain levels as well as ROM.

Line 384+ refers to ultrasound measurements of the shoulder. Ultrasound of the shoulder in our experience is very examiner dependent. Would all the technicians who carry out the ultrasound be all well trained in this technique?

Line 390  please define ‘full recovery’

Line 168 regarding the sample size: It is stated that 16 subjects are required in group one and 16 in group two to detect a difference equal or greater than 15 units in the SPADI. Would this mean that there are only eight patients in each group of 20ml and 40ml in both groups 1 and groups 2?

With a 20° loss to follow up of 20% then that leaves only six patients in each group. It seems unlikely that statistical significance between the groups could be obtained with such small numbers. This should be commented on. Also there should be a comment as to why the standard deviation for the SPADI is assumed to be 14pts. In our most recent study we found it was considerably more than this.

Discussion:

I would agree with the first paragraph that from the current literature it is uncertain as to the benefits of high volume and low volume hydrodilatation. Therefore, I do consider this study is well worthwhile carrying out.

It was earlier mentioned that in our department that at times capsular rupture occurs under 40mls of volume injected. The findings of reference 43 line 426-427 need to be better explained.

Line 433 states most studies show physiotherapy is useful complimentary treatment. There is only one reference, 46, given to this. From my knowledge of the literature there are actually no articles that have really demonstrated improvement with physiotherapy. There is certainly one paper which demonstrates no benefit of physiotherapy following hydrodilatation. It would be relevant and advantageous  if the results published concerning the benefits of physiotherapy was presented in more detail.

Regarding the comments on MRI line 443:

I really do not understand the statement that this test provides parameters that may be useful in cases with contradictory examination. This needs to be better explained.

Summary:

I do consider that given the current literature and understanding of capsulitis that this is a very worthwhile study, which could help in the future in improving the management for patients with capsulitis. In general it is fairly well set up. My main concern is the inclusion criteria where there is no mention of imaging being necessary to include patients in the study.

Comments on the Quality of English Language

Areas need to be improved, but overall good.

Author Response

Dear Reviewer,

We would like to sincerely thank you for your detailed and thoughtful review of our manuscript. Your comments have been extremely valuable in improving the scientific rigor, methodological clarity, and clinical relevance of our study. We greatly appreciate the time and expertise you have dedicated to evaluating our work.

Below, we provide point-by-point responses to each of your observations and suggestions.

Comment 1:

This is a very interesting, proposed trial. Regarding the current knowledge about adhesive capsulitis, it is very worthwhile and the results should be very helpful in the management of patients with capsulitis. The use of the word idiopathic will further clarify the study.

Response 1:

Dear reviewer, we are pleased to know that you enjoy the study, and we are working hard to address any questions you may have.

Comment 2:

The introduction was done well. I would agree with line 81 that the diagnosis of capsulitis is still to be developed. I would disagree with lines 89-91. There is no mention here of the value of plain x-ray or that glenohumeral joint arthritis may resemble capsulitis. There is only one reference cited for this statement which was rather obscure with no authors. I consider that for a patient presenting with a painful stiff shoulder that a plain x-ray is still a good baseline investigation. I do not see that ultrasound really has much place, and there is no reference cited to support this.

Response 2:

Regarding the introductory section regarding diagnostic testing, you are absolutely right that the need for such a basic test as an X-ray has not been mentioned. There is a reason for this. The study is performed in a specialist setting, and patients have already been filtered in Primary Care, where referral to our service requires a prior X-ray without data that could be associated with pathologies that associate symptoms like AC, such as those described in the "exclusion criteria" section on lines 144 and 159:

- Lines 93 – 100  “Currently, imaging tests focus on discarding pathologies that may simulate AC (rotator cuff tendinopathies, acromioclavicular osteoarthritis, labral injury, etc.) [2] X-ray, despite being a basic and necessary test, is utilized to discard bone pathologies. [14] Magnetic resonance imaging (MRI) and ultrasound (US) appear to discard soft tis-sue pathologies.

Although US is not currently widely used for AC, there is evidence that it is as re-liable as MRI in studying soft tissue parameters associated with AC [15,16]. "US can be incorporated into existing clinical diagnostic programs," concluded a meta-analysis on the use of US in the diagnosis of AC published in 2020 [17].”

2.Adhesive Capsulitis (Frozen Shoulder). Available online: https://www.ncbi.nlm.nih.gov/books/NBK532955/ (Accessed on 13 June 2024).

14. Vita, F.; Pederiva, D.; Tedeschi, R.; Spinnato, P.; Origlio, F.; Faldini, C. Adhesive capsulitis: the importance of early diagnosis and treatment. J Ultrasound. 2024, 27(3), 579–87.

15. Stella, S.M.; Gualtierotti, R.; Ciampi, B.; Trentanni, C.; Sconfienza, L.M.; Del Chiaro, A. Ultrasound features of adhesive capsulitis. Rheumatol Ther. 2022, 9(2), 481–95.

16. Al Khayyat, S.G.; Falsetti, P.; Conticini, E.; Frediani, B.; Galletti, S.; Stella, S.M. Adhesive capsulitis and ultrasound diagnosis, an inseparable pair: a novel review. J. Ultrasound 2022, 26(2), 369–384.

17. Wu, H.; Tian, H.; Dong, F.; Liang, W.; Song, D.; Zeng, J.; et al. The role of grey-scale ultrasound in the diagnosis of adhesive capsulitis of the shoulder: a systematic review and meta-analysis. Med. Ultrason. 2020, 22(3), 305.

Comment 3:

In line 95 it is stated that physical therapy may be useful in the management of capsulitis. To my knowledge there is no good study that has ever demonstrated this. Again Reference 2, which has no authors was cited.

Response 3:

Dear reviewer, the use of manual therapy in AC is a common practice both at a clinical - practical level and exposed in a multitude of studies and that is why it has been referenced with four articles (2, 13 -16). For the creation of this protocol, articles from 2024 to 2019 were reviewed and in many of them manual therapy is named as a first-line conservative treatment, being able to name articles used in the previous draft (14,28,42) and two new articles found in pubMED (51, 52).

The initial idea was to mention in the introduction which treatments are commonly used in the most recent publications. We did not want to elaborate on this aspect further, as we believe the focus should be on the HD technique itself. We will add new references in the new draft in the Discussion section, line 489, in favor of manual therapy, in order to justify its use as a complementary treatment.

14. Vita, F.; Pederiva, D.; Tedeschi, R.; Spinnato, P.; Origlio, F.; Faldini, C. Adhesive capsulitis: the importance of early diagnosis and treatment. J Ultrasound. 2024, 27(3), 579–87.

18. Hill, J.L. Evidence for Combining Conservative Treatments for AdhesiveCapsulitis. Ochsner Journal. 2024, 24(1), 47–52.

46. Whelan, G.; Yeowell, G.; Littlewood, C. Patient experiences of hydrodistension as a treatment for frozen shoulder: a longitudinal qualitative study. PLoS One. 2024, 19(6), e0304236.

50. Cho, C.H.; Bae, K.C.; Kim, D.H. Treatment strategy for frozen shoulder. Clin Orthop Surg. 2019, 11(3), 249.

51. Di Mascio, L.; Hamborg, T.; Mihaylova, B.; Kassam, J.; Shah, B.; Stuart, B.; et al. The Adhesive Capsulitis Cortico-steroid and Dilation (ACCorD) randomized controlled trial. Bone Jt. Open 2024, 5(3), 162–173.

52. Kayaokay, K.; Arslan Yurtlu, D. A comparison of the treatment outcomes with and without the use of intra-articular corticosteroids for frozen shoulder manipulation. Cureus 2023, 15(8), e44427.

Regarding reference 2, this belongs to the StatPearls platform, an online medical platform that offers educational and clinical review content, used primarily by healthcare professionals. Although it is not a primary source, its information is based on articles indexed in PubMed, making it a reliable source. As it is an online source, it has been referenced according to the criteria set forth by Clinical and Practice, so the authors do not appear in the reference.

Comment 4:

Concerning the materials and methods: In the inclusion criteria there is no mention of any imaging. It should be mentioned in the inclusion criteria that there should be no evidence of any osteoarthritis on any imaging. I consider a X-Ray should always be obtained in this situation.

Response 4:

Dear reviewer, this question has the same answer as question 2. The study is performed in a specialist setting, and patients have already been screened in Primary Care, where a prior X-ray is required for referral to our service.

To address this issue, we will add the following to lines 152 and 156:

Using X-rays and US, to discard the presence of conditions that can cause symptoms similar to AC, such as acromioclavicular osteoarthritis, labral injury, massive rotator cuff tear, or rheumatic diseases.”

Comment 5:

As mentioned later in the paper it is probably not cost effective to obtain an MRI scan in all cases. However, it could possibly be stated in the inclusion criteria that if there is any doubt as to the status of the glenohumeral joint on plain radiographs that an MRI scan would be obtained.

Response 5:

Dear reviewer, this research team considered, when creating the protocol, that the initial X-ray and the use of an ultrasound in the initial consultation to rule out soft tissue involvement would be necessary.

However, we believe your proposal may be appropriate for the completion of this study. Therefore, we propose adding the following paragraph to our protocol in the methods section, lines 283–286.

MRI request (11, 32): If, after treatment, ROM does not improve at 6 months after HD, according to the ROM recovery parameters for abduction, flexion ≥170°, and rotation ≥80°, described above, a MRI will be considered to discard other pathologies. This information will be reflected in the report and in the final results of the study.”

Additionally, this variable will be added to the page variables table.

This action would be supported by the following references, which subsequently support the use of MRI.

11. Picasso, R.; Pistoia, F.; Zaottini, F.; Marcenaro, G.; Miguel-Pérez, M.; Tagliafico, A.S. Adhesive Capsulitis of the Shoulder: Current Concepts on the Diagnostic Work-Up and Evidence-Based Protocol for Radiological Evaluation. Diagnostics. 2023, 13(22), 3410.

32. Dimitriou, D.; Winkler, E.; Zindel, C.; Grubhofer, F.; Wieser, K.; Bouaicha, S. Is routine magnetic resonance imaging necessary in patients with clinically diagnosed frozen shoulder? Utility of magnetic resonance im-aging in frozen shoulder. JSES Int. 2022, 6(5), 855–8.

Comment 6:

Line 141 exclusion: non- adherence to physiotherapy program – is this selection bias as little good literature to support physiotherapy as a successful treatment for AC.

Response 6:

Dear reviewer, as stated in comment 3, manual therapy after HD is a supported treatment. This criterion is included to maximize the equalization of the type of treatment received by all participants and thus avoid discrepancies between them. The only two factors we wish to differentiate are the developmental phase of the CA and the volumes used, so that this technique can be protocolized in the future. We believe that a loss of more than 20% of the pre-established sessions could cause confounding bias in the results obtained.

Comment 7:

There is also no exclusion of patients with a clinically proven full thickness cuff tear.

Response 7:

Dear reviewer, line 156 states the exclusion of massive rotator cuff tendon tears. We add, as we agree with you, the need to include full-thickness tears in lines 155–156.

Comment 8:

It is stated that differentiation as to whether pain or stiffness is a predominant feature when assigning the patients to stage 1 or 2. Is this based on the patient’s physical assessment or on the recorded VAS and ROM?

Response 8:

Dear reviewer,

Staging will be based on the patient's clinical evaluation, as described in lines 62–65. In this case, no cutoff points have been established for VAS or ROM. To clarify this staging, we have rewritten lines 175–181 as follows:

Patients in stage 1 will be considered those whose predominant clinical manifestation is nocturnal pain accompanied by progressive limitation of ROM; and patients in stage 2, those in whom there is a limitation of ROM and a reduction in pain, in relation to the previous months.”

Comment 9:

In line 190 it is stated that there is topping up the predetermined volume with saline solution later. Figure 3 is quoted, however the illustration for this appears to be figure 2. So I gather from figure 4 that the predetermined volume of saline solution was either 20ml or 40ml. In our department the hydrodilatations are done under fluoroscopy. We have found that in a number of cases of severe capsulitis that it is not possible to inject 40ml into the glenohumeral joint. Also some capsules actually rupture through the subscapularis bursa before 40mls has been injected. It needs to be stated how these cases with a very contracted capsule, which allow less than 40ml, would be dealt with under the protocol.

Response 9:

Dear reviewer, The reference to Figure 3 on line 212 refers to the HD technique performed using posterior approach ultrasound. Figure 2, referenced on line 203, corresponds to the suprascapular nerve block technique. The preset volumes are those previously listed on lines 199 and 200.

Regarding the question raised, the case you are describing is explained in the "Failed Technique" section on line 220, which is an exclusion criterion from the study and will be reflected in the descriptive results. This decision was made to be faithful to the random treatment assignment. I am leaving the "Failed Technique" section here, hoping you enjoy it.

Failed technique.

The HD technique can sometimes be difficult and may not be performed as previously described. For this reason, there are criteria by which the technique is considered unsuccessful:

- Acute pain that prevents the procedure from continuing at the patient's request.

- Inability to dilate the posterior joint capsule due to fluid leakage into soft tissue.

- Collapse of the joint capsule.

Comment 10:

It is then stated in line 191 that there is then a release manipulation. I gather that this is carried out while the local anesthetic is still within the joint, but this needs to be stated. Is this for both 20 and 40 Mls.

Response 10:

Dear reviewer, we agree with your comment. We have expanded the paragraph from lines 213 to 218, which reads as follows:

  • Finally, while the anesthetic is still having effect and after HD, manipulations will be performed in both groups of patients using release techniques based on Kaltenborn mobilization and the soft tissue energy technique, performing it passively without forcing the pain, meaning that it will be stopped at the moment the patient feels pain.”.

Comment 11:

I note the comments regarding the failed technique. I note in line 199 that one of these is inability to dilate the capsule due to fluid leakage into the soft tissue. This could be related not to a full thickness tearing of the rotator cuff, but a poor rotator cuff which inadequately maintains the fluid in the glenohumeral joint and allows it to leak into the subacromial bursa. Would these cases be still included under the current study protocol, or would, as stated, the technique then be considered unsuccessful and these patients be discarded from the study?

Response 11:

Again, regarding the exclusion criterion for "failed technique," our study does not specify the reasons for leaks, since loss of joint fluid would not make HD optimal and would therefore be a confounding factor.

In short, these patients would be excluded.

Comment 12:

Line 292, who is diagnosing phase 1 and Phase 2 and assigning the patient.

Response 12:

Dear reviewer,

Dr. Javier Muñoz Paz will classify patients into their stages, as explained in lines 254–257:“1st Recruitment, patient assignment and interventional treatment phase (Javier Muñoz Paz(J.M.-P)): The recruitment process will be conducted by J.M.-P during the initial consultation, who will screen and classify patients referred to the PM&R service based on the criteria outlined above

The person who performs the action is also represented in the flowchart on line 328.

Comment 13:

Line 346. Is the Lattinen Index for chronic pain an appropriate for assessment for patients’ pain level in AC given the progression and relatively short time range of phase 1?

Response 13:

Dear reviewer, thank you very much for raising this issue, which we consider to be of utmost importance.

The Lattinen Index is an increasingly used tool in the assessment of chronic pain. (39) It is true that it is not common to find its use currently in chronic shoulder pathology that associates pain as may be the case with adhesive capsulitis. However, this research team believes that it can provide relevant information such as the consumption of analgesics, an issue that is not taken into account with the EVA or the SPADI.

Despite this, this variable is a secondary variable and is used to homogenize the studies that are part of the doctoral thesis of Dr. Javier Muñoz Paz, which develops the subject “Second-line interventional treatments in chronic rotator cuff pathologies”.

We believe this contribution could also be made with the use of the EVA, so we will take note of it when discussing the final results of this trial, as it may be an influential factor in patient outcomes in Phase 1.

39. González, J.R.; Camba, A.; Muriel, C.; Rodríguez, M.; Contreras, D.; Barutell, C. Validación del índice de Lattinen para la evaluación del paciente con dolor crónico. Rev Soc Esp Dolor. 2012, 19(4), 181–8.

Comment 14:

In lines 363 and 375 there are two improvement scales utilized. I found it difficult to determine the difference between these two scales as both of them involved a single question, and the difference between the two needs to be better explained. Neither seems to address or specify pain levels as well as ROM.

Response 14:

Dear Reviewer,

Both the Patient Global Impression of Improvement Scale (PGI-I) and the Clinical Global Impression of Global Improvement Scale (CGI-GI) are scales used in chronic pain conditions. (40) Their purpose is to understand the patient's perception, in the case of PGI-I, and the specialist's perception, in the case of CGI-CI, of the results obtained after treatment. It is therefore a subjective variable, unrelated to the measurement of ROM, which is an objective variable.

Although these scales are not used in CA, this research team believes that their use as secondary variables can provide useful information regarding the comparison of the initial expectations of patients and/or the specialist at the end of treatment.

40. Sánchez, J.; Tejedor, A.; Carrascal, R. Atención al Paciente con Dolor Crónico No Oncológico en AP; Sociedad Española de Médicos Generales y de Familia en AP: Madrid, Spain, 2023; pp. 69–70.

Comment 15:

Line 384+ refers to ultrasound measurements of the shoulder. Ultrasound of the shoulder in our experience is very examiner dependent. Would all the technicians who carry out the ultrasound be all well trained in this technique?

Response 15:

Regarding the question raised, Dr. Javier Muñoz will perform the axillary recess (AR) measurement only during the initial consultation. Dr. Javier Muñoz Paz has three to four years of experience in diagnostic and interventional ultrasound of the rotator cuff, as well as a Master's degree in Musculoskeletal Ultrasound and Ultrasound-Guided Interventional Ultrasound. This supports his ultrasound expertise not only for AR measurement but also for performing the HD technique.

We believe it is important for this information to be clear in the protocol, so we will add the following information on line 267: “J.M.-P. will be responsible for these actions.” And on line 425: “The AR measurement will be performed only by J.M.-P. during the initial consultation.”

Comment 16:

Line 390 please define ‘full recovery’

Response 16:

Dear reviewer,

Thank you for your comment. We agree that we need to be clearer with this information, so we will replace lines 430 and 434 with the following information, which we hope will address any concerns.

Recovery of ROM for Abduction, Flexion ≥170° and ≥80° in rotations.”

Comment 17:

Line 168 regarding the sample size: It is stated that 16 subjects are required in group one and 16 in group two to detect a difference equal or greater than 15 units in the SPADI. Would this mean that there are only eight patients in each group of 20ml and 40ml in both groups 1 and groups 2? With a 20° loss to follow up of 20% then that leaves only six patients in each group. It seems unlikely that statistical significance between the groups could be obtained with such small numbers. This should be commented on. Also there should be a comment as to why the standard deviation for the SPADI is assumed to be 14pts. In our most recent study we found it was considerably more than this.

Response 17:

Dear reviewer,

Regarding the first question raised about sample size, the calculated sample size refers to each group based on the developmental stage. To clarify this aspect, we will modify the sample size paragraph with the following information.

Assuming an alpha risk of 0.05 and a statistical power greater than 0.8 in a bilateral contrast, 16 subjects in the 20 ml hydrodilation group and 16 in the 40 ml hydrodilation group are required to detect a difference equal to or greater than 15 units on the SPADI. Since stratification is available, the baseline size for each phase is therefore n=32 patients.”

In response to the second question, regarding the parameters used for sample size, we used parameters from the SPADI scale, a common scale for this type of pathology (30) that are reflected in other articles such as the one referenced in this case, in order to standardize the studies. (29)

We will add a new reference to provide greater consistency in the use of this scale.

29. Paruthikunnan, S.M.; Shastry, P.N.; Kadavigere, R.; Pandey, V.; Karegowda, L.H. Intra-articular steroid for adhesive capsulitis: does hydrodilatation give any additional benefit? A randomized control trial. Skeletal Radiol. 2020, 49(5), 795–803.

30. Parashar, A.; Goni, V.; Neradi, D.; Guled, U.; Rangasamy, K.; Batra, Y.K. Comparing three modalities of treatment for frozen shoulder: a prospective, double-blinded, randomized control trial. Indian J. Orthop. 2021, 55(2), 449–456.

Comment 18:

Discussion: I would agree with the first paragraph that from the current literature it is uncertain as to the benefits of high volume and low volume hydrodilatation. Therefore, I do consider this study is well worthwhile carrying out.

Response 18:

Dear reviewer, thank you very much for this comment.

This team believes it is necessary to standardize a technique that yields very good results in clinical practice and that only requires quality research studies for this purpose.

Comment 19:

It was earlier mentioned that in our department that at times capsular rupture occurs under 40mls of volume injected. The findings of reference 43 line 426-427 need to be better explained.

Response 19:

Dear reviewer, in order to resolve the doubts regarding the question raised, we have redeveloped the paragraph named between lines 462 - 482

The first of these differences concerns the volume required to achieve optimal capsular expansion. Reviewing the literature, we found that the volumes used in different studies vary greatly. In 2020, Jang Hyuk Cho concluded that the volume required to achieve optimal expansion was around 18 ml [44]. This same volume parameter, or similar values, also appeared in later studies on HD [14,45]. However, more up-to-date studies began to report HD with higher volumes, typically 30–40 ml [46]. In 2023, Sofia Dimitri-Pinheiro reported that her study used volumes close to 50 ml [43], as did the study by Magdalena Pimenta et al., which reported that a volume of up to 47 ml [47] allowed an optimal capsular expansion. These last two studies and some others agree that volumes greater than the initial 18 ml did not pose a risk of capsular rupture, a finding associated with worse outcomes [43,44,47].

This difference in volumes is the main reason for deciding to compare 20 ml vs. 40 ml in HD in this clinical trial.”

14. Vita, F.; Pederiva, D.; Tedeschi, R.; Spinnato, P.; Origlio, F.; Faldini, C. Adhesive capsulitis: the importance of early diagnosis and treatment. J Ultrasound. 2024, 27(3), 579–87.

43. Dimitri-Pinheiro, S.; Klontzas, M.E.; Vassalou, E.E.; Pimenta, M.; Soares, R.; Karantanas, A.H. Long-term outcomes of ultrasound-guided hydrodistension for adhesive capsulitis: a prospective observational study. Tomography. 2023, 9(5), 1857–67.

44. Cho, J.H. Updates on the treatment of adhesive capsulitis with hydraulic distension. Yeungnam Univ J Med. 2021, 38(1), 19–26.

45. Swaroop, S.; Gupta, P.; Patnaik, S.; Reddy, S. Intra-articular steroid alone vs hydrodilatation with intraarticular steroid in frozen shoulder – a randomised control trial. Malays Orthop J. 2023, 17(1), 34–9.

46. Whelan, G.; Yeowell, G.; Littlewood, C. Patient experiences of hydrodistension as a treatment for frozen shoulder: a longitudinal qualitative study. PLoS One. 2024, 19(6), e0304236.

47. Pimenta, M.; Vassalou, E.E.; Klontzas, M.E.; Dimitri-Pinheiro, S.; Ramos, I.; Karantanas, A.H. Ultrasound-guided hydrodilatation for adhesive capsulitis: capsule-preserving versus capsule-rupturing technique. Skeletal Radiol. 2024, 53(2), 253–61.

Comment 20:

Line 433 states most studies show physiotherapy is useful complimentary treatment. There is only one reference, 46, given to this. From my knowledge of the literature there are actually no articles that have really demonstrated improvement with physiotherapy. There is certainly one paper which demonstrates no benefit of physiotherapy following hydrodilatation. It would be relevant and advantageous if the results published concerning the benefits of physiotherapy was presented in more detail.

Response 20:

Dear reviewer, this question is answered in response 3 to comment 3.

Comment 21:

Regarding the comments on MRI line 443: I really do not understand the statement that this test provides parameters that may be useful in cases with contradictory examination. This needs to be better explained.

Response 21:

Dear Reviewer,

The use of MRI in the diagnosis of AC should not be indicated initially.

To give greater consistency to these guidelines, we will refer in lines 506-508 to the protocol proposed by Riccardo Picasso et al. in 2023, which stated that “MRI should be considered in doubtful cases in order to reveal differential factors” (11) for diseases with similar clinical features, such as labral involvement.

11. Picasso, R.; Pistoia, F.; Zaottini, F.; Marcenaro, G.; Miguel-Pérez, M.; Tagliafico, A.S. Adhesive Capsulitis of the Shoulder: Current Concepts on the Diagnostic Work-Up and Evidence-Based Protocol for Radiological Evaluation. Diagnostics. 2023, 13(22), 3410.

Comment 22:

Summary: I do consider that given the current literature and understanding of capsulitis that this is a very worthwhile study, which could help in the future in improving the management for patients with capsulitis. In general it is fairly well set up. My main concern is the inclusion criteria where there is no mention of imaging being necessary to include patients in the study.

Response 22:

Dear reviewer,

We are pleased to know that you consider this study extremely important in terms of the possibility of achieving improvements in the treatment of this disease.

We hope that the responses provided are to your liking and address the concerns you have raised. Regarding your main concern, we provide here a summary of the proposed updates and new references added to improve our protocol based on the approach you propose.

  • Lines 90 to 100 “Currently, imaging tests focus on discarding pathologies that may simulate AC (rotator cuff tendinopathies, acromioclavicular osteoarthritis, labral injury, etc.) [2] X-ray, despite being a basic and necessary test, is utilized to discard bone pathologies. [14] Magnetic resonance imaging (MRI) and ultrasound (US) appear to discard soft tis-sue pathologies.

Although US is not currently widely used for AC, there is evidence that it is as re-liable as MRI in studying soft tissue parameters associated with AC [15,16]. "US can be incorporated into existing clinical diagnostic programs," concluded a meta-analysis on the use of US in the diagnosis of AC published in 2020 [17].”

  • Lines 152 to 156: “Using X-rays and US, to discard the presence of conditions that can cause symptoms similar to AC, such as acromioclavicular osteoarthritis, labral injury, massive rotator cuff tear, or rheumatic diseases.”

  • Lines 283 to 286: MRI request (11, 32): If, after treatment, ROM does not improve at 6 months after HD, according to the ROM recovery parameters for abduction, flexion ≥170°, and rotation ≥80°, described above, a MRI will be considered to discard other pathologies. This information will be reflected in the report and in the final results of the study.

Finally, the draft article has been completely revised to improve the English of the text as requested. We hope the changes made will improve its intelligibility.

Once again, we are grateful for your thorough and constructive review. We hope that our revisions and clarifications have adequately addressed your concerns and strengthened the manuscript. Please do not hesitate to contact us if any further information is needed or if additional clarification is required.

Reviewer 4 Report

Comments and Suggestions for Authors

Overall: The purpose of the study described in this protocol paper is to compare whether patients with adhesive capsulitis (AC), stratified by phase 1 and 2, who receive high-volume HD as treatment, achieve better outcomes compared to patients who receive low-volume HD. This study will examine the utilization of novel treatment modalities that have the potential of being effective in the treatment of patients with AC. Thus, results from this study will hold both scientific and clinical importance. However, some additional details regarding the methodology that will be utilized is required, and in particular, the statistical approaches being utilized to answer the study objectives.

Line 141: There appears to be a typo -- should be "non-attendance" (vs. "attendance"), correct?

Lines 171-172: Why was a 15 unit change chosen in the SPADI? Is it based on previous intervention research within the AC population? Also, is a 15 point change consistent with the minimal detectable chance (MDC) for the SPADI in the AC population?

178-192: Please indicate the qualifications/training of the individual(s) performing these invasive procedures.

Lines 191-192: What specific joint mobilizations will be utilized (e.g., distraction, anterior glide, posterior glide, etc.)? Will all joint mobilizations be limited to the glenohumeral joint, or will mobilizations be applied to the scapulothoracic joint as well?

Lines 208-223: Consider creating supplementary materials that display photos of these exercises (sort of like a home exercise program you would give a patient). This will help improve the practical application of the protocol among clinicians.

Lines 224-225: More specific details regarding the mobilization and muscle energy manual therapies that will be utilized is needed. Also, is the manual therapy it limited to these interventions, or are other manual techniques (soft tissue mobilization, cupping, dry needling, etc.) going to be applied as well?

Lines 232-234: Will groups be sex-matched and age-matched? If not, will this be controlled for statistically (see below)?

Lines 256-270: Please consider making the statistical analyses its own section. Please also consider organizing the Objectives (1.3) with the statistical analyses in a manner that allows for the reader to understand how the research question in each objective is being answered via those variables and statistical tests.

Lines 256-270: Please indicate how statistical assumptions will be checked, and if assumptions are violated, how the researchers will manage those variables in the various statistical analyses. Please indicate if any covariates will be utilized in the described models and how the researchers will identify relevant covariates.

Lines 256-264: How will the data be prepared for these analyses? With change/difference scores be calculated for each variable? Or will the mean values at each time point be utilized. Further, it seems as though a mixed (between-within) model is required for a priori analyses. The researchers should consider using a 2x4 (group x time) RM ANOVA to initially examine for an interaction effect, and then based on those results, conduct the aforementioned between group comparisons and within group comparisons at each time point. Additionally, given the variation in time between each data collection session (i.e., 1 month vs. 3 months), consider using some more advance linear modeling techniques that can account for this factor.

Lines 317-318: What is the MDC for the SPADI within the AC patient population?

Lines 428-434: Please consider discussing the fact that a proper control group is not being utilized (i.e., the study is not a randomized control trial). A control in this type of study could just be standard PT treatment without any HD (vs. only comparing low vs. high volume HD). This could be considered a limitation of the trial.

Figure 3: The caption for Figure 3 is incorrectly labeled as "Figure 2". Please correct.

Author Response

First of all, we would like to sincerely thank you for the time and effort dedicated to reviewing our manuscript. We truly appreciate your positive comment noting that our study is well organized and that the protocol is clearly presented.

The purpose of the study described in this protocol paper is to compare whether patients with adhesive capsulitis (AC), stratified by phase 1 and 2, who receive high-volume HD as treatment, achieve better outcomes compared to patients who receive low-volume HD. This study will examine the utilization of novel treatment modalities that have the potential of being effective in the treatment of patients with AC. Thus, results from this study will hold both scientific and clinical importance. However, some additional details regarding the methodology that will be utilized is required, and in particular, the statistical approaches being utilized to answer the study objectives.”

Line 141: There appears to be a typo -- should be "non-attendance" (vs. "attendance"), correct?

  • Estimated reviewer, we agree with that error and it has been corrected.

Lines 171-172: Why was a 15 unit change chosen in the SPADI? Is it based on previous intervention research within the AC population? Also, is a 15 point change consistent with the minimal detectable chance (MDC) for the SPADI in the AC population?

Lines 317-318: What is the MDC for the SPADI within the AC patient population?

  • Estimated reviewer, the parameters used to calculate the sample size were obtained from the referenced study, in which the SPADI scale was used to assess patients with AC who had received HD treatment.

29. Paruthikunnan, S.M.; Shastry, P.N.; Kadavigere, R.; Pandey, V.; Karegowda, L.H. Intra-articular steroid for adhesive capsulitis: does hydrodilatation give any additional benefit? A randomized control trial. Skeletal Radiol. 2020, 49(5), 795–803.

178-192: Please indicate the qualifications/training of the individual(s) performing these invasive procedures.

  • Regarding the question raised, Dr. Javier Muñoz Paz, a Physical Medicine and Rehabilitation specialist, has three to four years of experience in diagnostic and interventional ultrasound of the rotator cuff, as well as a Master's degree in Musculoskeletal Ultrasound and Ultrasound-Guided Interventional Ultrasound. This supports his expertise in the procedures described.

Lines 191-192: What specific joint mobilizations will be utilized (e.g., distraction, anterior glide, posterior glide, etc.)? Will all joint mobilizations be limited to the glenohumeral joint, or will mobilizations be applied to the scapulothoracic joint as well?

  • Estimated reviewer, patients will be treated according to the protocol set out between lines 227 - 251. In this protocol, it is stated that passive mobilization exercises, scapular mobilization, distraction exercises according to Kaltenborn mobilization and muscle energy mobilization will be performed.

Lines 208-223: Consider creating supplementary materials that display photos of these exercises (sort of like a home exercise program you would give a patient). This will help improve the practical application of the protocol among clinicians.

Lines 224-225: More specific details regarding the mobilization and muscle energy manual therapies that will be utilized is needed. Also, is the manual therapy it limited to these interventions, or are other manual techniques (soft tissue mobilization, cupping, dry needling, etc.) going to be applied as well?

  • Estimated reviewer, we welcome your feedback for future publications. In this article, the main focus is therefore on comparing the results that can be achieved with different volumes in HD, as well as the influence of phases. That's why the only images we considered including are those of the same technique, so as not to diminish its importance. We know the importance of manual therapy in these types of patients, and that's why we wanted to include it and give it its due space by developing a protocol, as mentioned above.

The techniques used are based on references such as those mentioned.

31. Pattnaik, S.; Kumar, P.; Sarkar, B.; Oraon, A.K. Comparison of Kaltenborn mobilization technique and muscle energy technique on range of motion, pain and function in subjects with chronic shoulder adhesive capsulitis. Hong Kong Physiother J. 2023, 43(2), 149–59.

Lines 232-234: Will groups be sex-matched and age-matched? If not, will this be controlled for statistically (see below)?

  • Estimated reviewer, this study will not be matched by age or sex, as the influence of age or sex on these outcomes is not being studied. Despite this, we find it interesting for future studies.

Lines 256-270: Please consider making the statistical analyses its own section. Please also consider organizing the Objectives (1.3) with the statistical analyses in a manner that allows for the reader to understand how the research question in each objective is being answered via those variables and statistical tests.

Lines 256-270: Please indicate how statistical assumptions will be checked, and if assumptions are violated, how the researchers will manage those variables in the various statistical analyses. Please indicate if any covariates will be utilized in the described models and how the researchers will identify relevant covariates.

Lines 256-264: How will the data be prepared for these analyses? With change/difference scores be calculated for each variable? Or will the mean values at each time point be utilized. Further, it seems as though a mixed (between-within) model is required for a priori analyses. The researchers should consider using a 2x4 (group x time) RM ANOVA to initially examine for an interaction effect, and then based on those results, conduct the aforementioned between group comparisons and within group comparisons at each time point. Additionally, given the variation in time between each data collection session (i.e., 1 month vs. 3 months), consider using some more advance linear modeling techniques that can account for this factor.

  • Estimated reviewer, regarding the aspects you propose regarding statistical methods, the organization of the article has followed the guidelines of the template included in Clinics and Practice, both in the placement of the statistical analysis section and the objectives.

In response to the question on lines 256-270, the purpose of the statistical study is to determine the normality of the sample distribution in order to use parametric tests to provide greater consistency to the statistical results. Otherwise, nonparametric tests should be used. For further explanation, we have added the following paragraph on lines 296-300:

Before performing statistical tests, the validity of the analyses will be assessed. The normality of the variables will be evaluated using the Shapiro-Wilk test. The homogeneity of variances between groups will be analyzed using the Levene test. In the case of regression models, the linearity and independence of the residuals will be verified using scatter plots and residual analysis.”

Although we believe, as you do, that the information that can be provided by using the covariates described in section 3.1 Variables is important, those not described in the section on primary and secondary objectives will be used only to obtain descriptive data.

The purpose of the study is to demonstrate whether the results of HD differ when modifying the volume considered optimal for performing a capsular expansion, in addition to the influence of the evolutionary phase.

We welcome the proposed idea, with a view to opening up various lines of research once the trial is completed that focus on the influence of these covariates, such as age, gender, illness, etc.

As a final response regarding data analysis, the initial idea would be to analyze the absolute values ​​of the main variables both longitudinally pre-post and cross-sectionally at each time point.

The tests chosen for statistical analysis are available to the statistician and the relevant studies discussed above in the newly added paragraph. We therefore welcome your suggestion and will consider it if the data can be analyzed using a 2x4 RM ANOVA or an advanced linear model.

Lines 428-434: Please consider discussing the fact that a proper control group is not being utilized (i.e., the study is not a randomized control trial). A control in this type of study could just be standard PT treatment without any HD (vs. only comparing low vs. high volume HD). This could be considered a limitation of the trial.

  • Estimated reviewer, this team considers the trial if it has all the elements to be defined as a “Randomized, controlled, triple-blind and prospective clinical trial” since the effect of HD at different volumes in AC patients is being evaluated, the patients will be randomly assigned to each treatment, two different interventions are compared (40 ml vs. 20 ml) that serve as a control group with each other and finally neither the clinical evaluator, nor the statistician nor the patient know this assignment.

I emphasize again that the purpose of this study is not to demonstrate improved outcomes between patients who receive HD versus those who do not, but rather to study whether HD outcomes differ when the volume considered optimal for capsular expansion is modified.

Figure 3: The caption for Figure 3 is incorrectly labeled as "Figure 2". Please correct.

  • Estimated reviewer. The reference to Figure 3 on line 212 refers to the HD technique performed using posterior approach ultrasound. Figure 2, referenced on line 203, corresponds to the suprascapular nerve block technique. We did not find the error indicated.

We hope these revisions appropriately address your recommendation and contribute to enhancing the scientific quality of the manuscript.

Once again, thank you for your valuable comments.

Round 2

Reviewer 3 Report

Comments and Suggestions for Authors

The authors have responded well to the reviewer's comments.

The paper is now much improved.

Author Response

We sincerely thank the reviewer for the time and effort dedicated to evaluating our manuscript and for the valuable comments that helped improve the quality of our work.

We are pleased to know that our responses were found appropriate and satisfactory.

We truly appreciate the reviewer’s thorough and constructive feedback

Reviewer 4 Report

Comments and Suggestions for Authors

The authors have addressed some of the lack of clarity regarding the methodological approaches, however, some fatal flaws still exist. Since this is a methods/protocol paper, the reviewer treated it as if the paper was a proposal (e.g., grant, thesis, dissertation, etc.), and as such, this reviewer would have some significant concerns in the proposed statistical methodology and interpretation of analyses.

Original Reviewer Comment:

Lines 171-172: Why was a 15 unit change chosen in the SPADI? Is it based on previous intervention research within the AC population? Also, is a 15 point change consistent with the minimal detectable chance (MDC) for the SPADI in the AC population?

Lines 317-318: What is the MDC for the SPADI within the AC patient population?

Author Response:

  • Estimated reviewer, the parameters used to calculate the sample size were obtained from the referenced study, in which the SPADI scale was used to assess patients with AC who had received HD treatment.

29. Paruthikunnan, S.M.; Shastry, P.N.; Kadavigere, R.; Pandey, V.; Karegowda, L.H. Intra-articular steroid for adhesive capsulitis: does hydrodilatation give any additional benefit? A randomized control trial. Skeletal Radiol. 2020, 49(5), 795–803.

It is understandable that the Paruthikunnan et al. (2020) paper would be utilized to calculate a sample size from a power analysis as it is another intervention study. However, there is a lack of clinical significance in the proposed assessment of the ultimate outcomes of the patients. Specifically, it is known from the scientific literature that the minimal detectable change (MDC) for the SPADI among patients with AC is 17.0 (Tveitå EK, et al., 2008). This indicates that a change in 17 points on the SPADI is required for it to be considered "real" change -- this is irrespective of any statistically significant change, as a change could statistically significant, but not clinically significant (and vice versa). Therefore, the overall assessment of outcomes among these patients should also be viewed from a clinical effectiveness lens -- both at a group level and at a patient level. If the mean change in SPADI scores does not exceed 17.0 in all patients, then this intervention was not meaningful for all patients, regardless of what the group-based bayesian statistical analyses suggest. As a clinician, this type of change in functional outcomes is paramount in determining the success of an overall intervention within a given patient. Thus, consideration to examining such changes within the randomized clinical trial should be taken, otherwise the clinical implications will not be as evident to the ultimate audience.

Tveitå EK, Ekeberg OM, Juel NG, Bautz-Holter E. Responsiveness of the shoulder pain and disability index in patients with adhesive capsulitis. BMC Musculoskelet Disord. 2008;9:161.   Original Reviewer Comment:

Lines 256-264: How will the data be prepared for these analyses? With change/difference scores be calculated for each variable? Or will the mean values at each time point be utilized. Further, it seems as though a mixed (between-within) model is required for a priori analyses. The researchers should consider using a 2x4 (group x time) RM ANOVA to initially examine for an interaction effect, and then based on those results, conduct the aforementioned between group comparisons and within group comparisons at each time point. Additionally, given the variation in time between each data collection session (i.e., 1 month vs. 3 months), consider using some more advance linear modeling techniques that can account for this factor.

Author Response:

  • Estimated reviewer, regarding the aspects you propose regarding statistical methods, the organization of the article has followed the guidelines of the template included in Clinics and Practice, both in the placement of the statistical analysis section and the objectives.

In response to the question on lines 256-270, the purpose of the statistical study is to determine the normality of the sample distribution in order to use parametric tests to provide greater consistency to the statistical results. Otherwise, nonparametric tests should be used. For further explanation, we have added the following paragraph on lines 296-300:

Before performing statistical tests, the validity of the analyses will be assessed. The normality of the variables will be evaluated using the Shapiro-Wilk test. The homogeneity of variances between groups will be analyzed using the Levene test. In the case of regression models, the linearity and independence of the residuals will be verified using scatter plots and residual analysis.”

Although we believe, as you do, that the information that can be provided by using the covariates described in section 3.1 Variables is important, those not described in the section on primary and secondary objectives will be used only to obtain descriptive data.

The purpose of the study is to demonstrate whether the results of HD differ when modifying the volume considered optimal for performing a capsular expansion, in addition to the influence of the evolutionary phase.

We welcome the proposed idea, with a view to opening up various lines of research once the trial is completed that focus on the influence of these covariates, such as age, gender, illness, etc.

As a final response regarding data analysis, the initial idea would be to analyze the absolute values ​​of the main variables both longitudinally pre-post and cross-sectionally at each time point.

The tests chosen for statistical analysis are available to the statistician and the relevant studies discussed above in the newly added paragraph. We therefore welcome your suggestion and will consider it if the data can be analyzed using a 2x4 RM ANOVA or an advanced linear model.

Thank you for addressing the assumptions of normality and homogeneity of variances. Even if the assumption of normality is violated, the authors should consider performing transformations on the data to normalize the distribution, as general linear models are going to provide a far more robust data analysis than the non-parametric analyses. The described non-parametric analyses should only be utilized as a last resort when examining RCT outcomes.

Further, analysis of any main effects or between group effects without first performing an omnibus test examining for a group x time interaction would be inappropriate, as it could present the results inappropriately to the reader. For example, if a significant 2 x 4 (HD technique x time) interaction effect is identified, this group x time interaction indicates that the way time affects a given outcome measure does in fact depend on HD technique, in which case the originally described between (independent t-tests) and within (paired t-tests) analyses should be performed in a post hoc follow-up fashion. However, interpretation of all the actual omnibus tests of the multifactorial within-between 2 x 4 RM ANOVA is still important.

In addition, the researchers should consider examining for group differences at baseline, as if group differences exist, these differences will need to be accounted for in the associated statistical modeling (e.g., covariate, calculate of change metric, etc.).

Finally, although the hope is that the model is ultimately properly powered, given the small sample size, and the desire to identify HD technique on patient outcomes, the researchers should consider calculating effect sizes to provide further description of the observed treatment effects, particularly if this proposed study is a pilot study, or if non-significant findings are observed, as it could suggest sample size limitations are masking a true significant effect due to low observed power.

Original Reviewer Comment:

Lines 428-434: Please consider discussing the fact that a proper control group is not being utilized (i.e., the study is not a randomized control trial). A control in this type of study could just be standard PT treatment without any HD (vs. only comparing low vs. high volume HD). This could be considered a limitation of the trial.

Author Response:

  • Estimated reviewer, this team considers the trial if it has all the elements to be defined as a “Randomized, controlled, triple-blind and prospective clinical trial” since the effect of HD at different volumes in AC patients is being evaluated, the patients will be randomly assigned to each treatment, two different interventions are compared (40 ml vs. 20 ml) that serve as a control group with each other and finally neither the clinical evaluator, nor the statistician nor the patient know this assignment.

Yes, this proposed study is certainly a randomized trial, but it could be argued that a true control group is not being utilized. For example, the researchers may observed that the 40 ml HD technique yielded better outcomes than the 20 ml HD technique, but it is unknown if any of these techniques are better than no formal treatment at all. This is particularly relevant for adhesive capsulitis, as it is often simply self-limiting and resolves without formal medical or rehabilitative intervention. Thus, this study design is examining the effectiveness of HD techniques compared to each other, but it is not examining the effectiveness of the HD techniques to a proper control group. It is understandable that this study may simply be a pilot study, and a more formal control group may be hypothetically included in future large-scale studies, but recognition of this limitation is still warranted.

Original Reviewer Comment:

Figure 3: The caption for Figure 3 is incorrectly labeled as "Figure 2". Please correct.

Author Response:

  • Estimated reviewer. The reference to Figure 3 on line 212 refers to the HD technique performed using posterior approach ultrasound. Figure 2, referenced on line 203, corresponds to the suprascapular nerve block technique. We did not find the error indicated.

Figure 3 caption remains incorrectly labeled.  Specifically, it reads: "Figure 2 – Illustration of access point for posterior approach HD technique."  The photos themselves are labeled correctly (3a, 3b, 3c), but the caption is labeled incorrectly.

Author Response

We once again thank you for the time and effort you put into reviewing our manuscript. We appreciate your numerous comments, knowing that the ultimate goal is to improve our manuscript and achieve greater scientific dissemination.

-----------------------------------------------------------------------------------------------------------------

“Comments and Suggestions for Authors

The authors have addressed some of the lack of clarity regarding the methodological approaches, however, some fatal flaws still exist. Since this is a methods/protocol paper, the reviewer treated it as if the paper was a proposal (e.g., grant, thesis, dissertation, etc.), and as such, this reviewer would have some significant concerns in the proposed statistical methodology and interpretation of analyses.”

New comment regarding the first proposal

Lines 171-172: Why was a 15 unit change chosen in the SPADI? Is it based on previous intervention research within the AC population? Also, is a 15 point change consistent with the minimal detectable chance (MDC) for the SPADI in the AC population?

Lines 317-318: What is the MDC for the SPADI within the AC patient population?

It is understandable that the Paruthikunnan et al. (2020) paper would be utilized to calculate a sample size from a power analysis as it is another intervention study. However, there is a lack of clinical significance in the proposed assessment of the ultimate outcomes of the patients. Specifically, it is known from the scientific literature that the minimal detectable change (MDC) for the SPADI among patients with AC is 17.0 (Tveitå EK, et al., 2008). This indicates that a change in 17 points on the SPADI is required for it to be considered "real" change -- this is irrespective of any statistically significant change, as a change could statistically significant, but not clinically significant (and vice versa). Therefore, the overall assessment of outcomes among these patients should also be viewed from a clinical effectiveness lens -- both at a group level and at a patient level. If the mean change in SPADI scores does not exceed 17.0 in all patients, then this intervention was not meaningful for all patients, regardless of what the group-based bayesian statistical analyses suggest. As a clinician, this type of change in functional outcomes is paramount in determining the success of an overall intervention within a given patient. Thus, consideration to examining such changes within the randomized clinical trial should be taken, otherwise the clinical implications will not be as evident to the ultimate audience.

Tveitå EK, Ekeberg OM, Juel NG, Bautz-Holter E. Responsiveness of the shoulder pain and disability index in patients with adhesive capsulitis. BMC Musculoskelet Disord. 2008;9:161.

Answer: Estimated reviewer, this article was not previously referenced because, as this project is part of a doctoral thesis, we sought to reference articles that were within the past five years,  as far as it has been possible. Despite this, we consider your proposal to be very specific and useful for improving our project, as it is essential to understand the two concepts you present. It is of no use to us to have significant results if no minimal clinical improvement is observed in clinical practice.

To resolve this issue, the sample size has been recalculated according to both articles (the one proposed by you and ours) lines 176 - 182 and the following information will be added to the SPADI variable with its corresponding reference:

“It has been shown that the minimal clinically important difference (MCID) for changes in SPADI should be ≥ 17 points.”

  1. Tveitå, E.K.; Ekeberg, O.M.; Juel, N.G.; Bautz-Holter, E. Responsiveness of the shoulder pain and disability index in patients with adhesive capsulitis. BMC Musculoskelet. Disord. 2008, 9, 161. 

New comment regarding the first proposal

Lines 256-264: How will the data be prepared for these analyses? With change/difference scores be calculated for each variable? Or will the mean values at each time point be utilized. Further, it seems as though a mixed (between-within) model is required for a priori analyses. The researchers should consider using a 2x4 (group x time) RM ANOVA to initially examine for an interaction effect, and then based on those results, conduct the aforementioned between group comparisons and within group comparisons at each time point. Additionally, given the variation in time between each data collection session (i.e., 1 month vs. 3 months), consider using some more advance linear modeling techniques that can account for this factor.

Thank you for addressing the assumptions of normality and homogeneity of variances. Even if the assumption of normality is violated, the authors should consider performing transformations on the data to normalize the distribution, as general linear models are going to provide a far more robust data analysis than the non-parametric analyses. The described non-parametric analyses should only be utilized as a last resort when examining RCT outcomes.

Further, analysis of any main effects or between group effects without first performing an omnibus test examining for a group x time interaction would be inappropriate, as it could present the results inappropriately to the reader. For example, if a significant 2 x 4 (HD technique x time) interaction effect is identified, this group x time interaction indicates that the way time affects a given outcome measure does in fact depend on HD technique, in which case the originally described between (independent t-tests) and within (paired t-tests) analyses should be performed in a post hoc follow-up fashion. However, interpretation of all the actual omnibus tests of the multifactorial within-between 2 x 4 RM ANOVA is still important.

In addition, the researchers should consider examining for group differences at baseline, as if group differences exist, these differences will need to be accounted for in the associated statistical modeling (e.g., covariate, calculate of change metric, etc.).

Finally, although the hope is that the model is ultimately properly powered, given the small sample size, and the desire to identify HD technique on patient outcomes, the researchers should consider calculating effect sizes to provide further description of the observed treatment effects, particularly if this proposed study is a pilot study, or if non-significant findings are observed, as it could suggest sample size limitations are masking a true significant effect due to low observed power.

 Answer:   Regarding the first paragraph, the fundamental idea of ​​this study is to seek results based on parametric tests, leaving the use of non-parametric tests for last resort, as you propose.

Regarding the need to analyze the interaction of time with our results, our current sample size (32 patients total per group, 64 patients across the two phases) could be calculated using a 2x4 ANOVA with a medium effect size (f=0.25).

All statistical studies carried out will have a prior study to determine the homogeneity of the samples, in order to determine whether there are differences between them, as already explained in line 286.

Finally, regarding the "small" sample size you mention, the sample size calculations were obtained using GRANMO (28) based on studies referenced in the article, including the one you proposed (29, 31). Both the aspect of significant results and the MDIC of the main variables (SPADI ≥17 points and EVA ≥ 2) were taken into account. Therefore, this size should not be a problem, as long as we can reach the target of 32 patients per phase, a minimum of 64 patients in total.

In addition, we also want to emphasize that all statistical studies will be conducted by an external expert statistician, who will be allowed to obtain all relevant information so that the results can be analyzed and discussed as effectively as possible in the future.

Therefore, the following information will be added to the “4th Phase of Statistical Analysis” section on lines 350 – 351:

“A 2 × 4 mixed repeated measures ANOVA will be performed to examine the group × time interaction on the outcome variables, in order to determine whether the time course differs between the analyzed techniques.”

  1. GRANMO. Available from: https://www.datarus.eu/aplicaciones/granmo/ (Accessed on 17 August 2024).
  2. Paruthikunnan, S.M.; Shastry, P.N.; Kadavigere, R.; Pandey, V.; Karegowda, L.H. Intra-articular steroid for adhesive capsulitis: does hydrodilatation give any additional benefit? A randomized control trial. Skeletal Radiol. 2020, 49(5), 795–803.
  3. Tveitå, E.K.; Ekeberg, O.M.; Juel, N.G.; Bautz-Holter, E. Responsiveness of the shoulder pain and disability index in patients with adhesive capsulitis. BMC Musculoskelet. Disord. 2008, 9, 161.

New comment regarding the first proposal

Lines 428-434: Please consider discussing the fact that a proper control group is not being utilized (i.e., the study is not a randomized control trial). A control in this type of study could just be standard PT treatment without any HD (vs. only comparing low vs. high volume HD). This could be considered a limitation of the trial.

Yes, this proposed study is certainly a randomized trial, but it could be argued that a true control group is not being utilized. For example, the researchers may observed that the 40 ml HD technique yielded better outcomes than the 20 ml HD technique, but it is unknown if any of these techniques are better than no formal treatment at all. This is particularly relevant for adhesive capsulitis, as it is often simply self-limiting and resolves without formal medical or rehabilitative intervention. Thus, this study design is examining the effectiveness of HD techniques compared to each other, but it is not examining the effectiveness of the HD techniques to a proper control group. It is understandable that this study may simply be a pilot study, and a more formal control group may be hypothetically included in future large-scale studies, but recognition of this limitation is still warranted.

 Answer: Tal y como usted comenta, este ensayo clínico está realizado para determinar si la HD con 40ml de volumen, consigue mejores resultados con respecto a la HD con 20 ml, ya que ese es el objetivo principal que se expresa en el borrador.

“Patients with AC, depending on the stage of progression, who receive high-volume HD as treatment, obtain better results on the Shoulder Pain and Disability Index (SPADI), the Visual Analog Scale (VAS), and ROM at the first, third, and sixth months of therapy, compared to patients receiving low-volume HD in the general population.”

We understand the idea you're trying to convey, and like you, we believe it's an interesting topic for discussion. However, the creation of a third branch without our involvement would mean:

  • Adopting an attitude, known and described in our article as "benign negligence," by which AC resolves in a limited way. This attitude has allowed us to know that patients usually have an average duration of AC of approximately 2-3 years until resolution. (13)
  • Putting the focus and objective on comparing the usefulness of HD + FST vs. “benign negligence” is an issue that is not raised in our study and, reinforcing this idea, our objective, as you mention, is focused on “examining the effectiveness of HD techniques, compared to each other.”
  • To address the question you propose, a completely different clinical trial would have to be conducted.

That said, this research team believes that the question you raise could be of interest for the discussion of the final results, but we  do not consider the lack of a third arm without any treatment to be a limitation for achieving the objectives set in our study.

  1. Ammerman, B.M.; Dennis, E.R.; Ling, D.; Hannafin, J.A. Ultrasound-Guided Glenohumeral Corticosteroid Injection for the Treatment of Adhesive Capsulitis of the Shoulder: The Role of Clinical Stage in Response to Treatment. Sports Health. 2024, 16(3), 333–9.

New comment regarding the first proposal

Figure 3: The caption for Figure 3 is incorrectly labeled as "Figure 2". Please correct.

Figure 3 caption remains incorrectly labeled.  Specifically, it reads: "Figure 2 – Illustration of access point for posterior approach HD technique."  The photos themselves are labeled correctly (3a, 3b, 3c), but the caption is labeled incorrectly.

Answer:Thank you very much for explaining the error. We noticed that in our Word document, the numbering was "Figure 3," and when we converted it to PDF, it changed to "Figure 2." We've now resolved this issue.

 We hope that all the responses are to your liking and we thank you, once again, for the interest you have shown in improving our project.
